# A small-molecule inhibitor of SOD1-Derlin-1 interaction ameliorates pathology in an ALS mouse model

Naomi Tsuburaya[1], Kengo Homma [1], Tsunehiko Higuchi[2], Andrii Balia[3], Hiroyuki Yamakoshi[2],
Norio Shibata [3], Seiichi Nakamura[2], Hidehiko Nakagawa[2], Shin-ichi Ikeda[2], Naoki Umezawa[2], Nobuki Kato[2],
Satoshi Yokoshima[4], Masatoshi Shibuya[4], Manabu Shimonishi[5], Hirotatsu Kojima[6], Takayoshi Okabe[6],
Tetsuo Nagano[6], Isao Naguro[1], Keiko Imamura[7,8,9], Haruhisa Inoue[7,8,9], Takao Fujisawa[1] & Hidenori Ichijo[1,6]

Amyotrophic lateral sclerosis (ALS) is a devastating neurodegenerative disorder. Despite its severity, there are no effective treatments because of the complexity of its pathogenesis. As one of the underlying mechanisms of *Cu, Zn superoxide dismutase* (*SOD1*) gene mutation-induced ALS, SOD1 mutants (SOD1$^{mut}$) commonly interact with an endoplasmic reticulum-resident membrane protein Derlin-1, triggering motoneuron death. However, the importance of SOD1-Derlin-1 interaction in in vitro human model and in vivo mouse model remains to be elucidated. Here, we identify small-molecular-weight compounds that inhibit the SOD1-Derlin-1 interaction by screening approximately 160,000 compounds. The inhibitor prevents 122 types of SOD1$^{mut}$ from interacting with Derlin-1, and significantly ameliorates the ALS pathology both in motoneurons derived from patient induced pluripotent stem cells and in model mice. Our data suggest that the SOD1-Derlin-1 interaction contributes to the pathogenesis of ALS and is a promising drug target for ALS treatment.

[1] Laboratory of Cell Signaling, Graduate School of Pharmaceutical Sciences, The University of Tokyo, 7-3-1 Hongo, Bunkyo-ku, Tokyo 113-0033, Japan.
[2] Graduate School of Pharmaceutical Sciences, Nagoya City University, 3-1 Tanabe-dori, Mizuho-ku, Nagoya 467-8603, Japan. [3] Department of Life Science and Applied Chemistry, Nagoya Institute of Technology, Gokiso, Showa-ku, Nagoya 466-8555, Japan. [4] Graduate School of Pharmaceutical Sciences, Nagoya University, Furo-cho, Chikusa-ku, Nagoya 464-8601, Japan. [5] GCOE Program, The University of Tokyo, 7-3-1 Hongo, Bunkyo-ku, Tokyo 113-0033, Japan. [6] Drug Discovery Initiative (DDI), The University of Tokyo, 7-3-1 Hongo, Bunkyo-ku, Tokyo 113-0033, Japan. [7] Center for iPS Cell Research and Application (CiRA), Kyoto University, Kyoto 606-8507, Japan. [8] iPSC-based Drug Discovery and Development Team, RIKEN BioResource Center, Kyoto 606-8507, Japan. [9] Medical-risk Avoidance based on iPS Cells Team, RIKEN Center for Advanced Intelligence Project (RIKEN AIP), Kyoto 606-8507, Japan. These authors contributed equally: Naomi Tsuburaya, Kengo Homma.  Correspondence and requests for materials should be addressed to H.I. (email: ichijo@mol.f.u-tokyo.ac.jp)

Amyotrophic lateral sclerosis (ALS) is a progressive, late-onset neurodegenerative disorder characterized by the selective loss of both upper and lower motoneurons[1]. Riluzole and Edaravone are the only two available treatments for ALS that provide a modest improvement in survival[2–4]. Although some clinical trials have been conducted over the past 20 years, most of them have failed, at least in part because of the poor understanding of the pathogenesis of ALS. Multiple motoneurons toxicities have been proposed, such as oxidative stress, excitotoxicity, proteasome dysfunction, endoplasmic reticulum (ER) stress, abnormal mitochondrial function, and altered axonal transport, but no consensus or precise mechanisms have been elucidated[5]. Thus, there remains strong demand for finding new and effective ALS treatments based on the molecular mechanisms of ALS pathogenesis.

Mutations in the *Cu, Zn superoxide dismutase* (*SOD1*) gene are frequently found in familial ALS (FALS) patients[6]. Although SOD1 is a well-known antioxidant enzyme, mutant SOD1 (SOD1$^{mut}$)-mediated FALS is now considered to result from gain-of-toxic function(s) of SOD1 by mutation rather than from changes in superoxide dismutase activity[7–12]. To date, more than 170 different mutations have been reported, but only a few reports have tested the commonality of toxic mechanism(s) by using a substantial number of different ALS-related SOD1$^{mut}$. Therefore, further investigation of SOD1$^{mut}$-induced toxic mechanism(s) is required for managing this fatal neurodegenerative disease.

We have previously reported that over 100 types of ALS-related SOD1$^{mut}$ but not wild-type SOD1 (SOD1$^{WT}$) commonly expose the Derlin-1-binding region (termed the DBR) and interact via the DBR with Derlin-1, one of the critical components of the ER-associated degradation (ERAD) machinery[13,14]. Moreover, disruption of the SOD1$^{mut}$-Derlin-1 interaction by expressing Derlin-1-derived peptide (termed Derlin-1 (CT4)), which corresponds to the binding site for the DBR and competes for binding to SOD1$^{mut}$, has been shown to be protective against SOD1$^{mut}$-dependent motoneuron loss in the mouse primary spinal cord culture[13]. These findings suggest that the SOD1$^{mut}$-Derlin-1 interaction is a common feature of SOD1$^{mut}$-caused FALS pathology and that inhibition of this interaction may be a potential target for ALS treatment.

Protein–protein interaction (PPI) is an attractive target for drug development because it plays central roles in large parts of biological processes, and in pathological conditions. Thus, small molecules that inhibit PPIs have potential as therapeutic medicines themselves and also as useful experimental tools to elucidate pathological mechanisms. Although it is considered as challenging to identify small molecules that modulate PPIs, several reports provide encouraging evidence for finding such compounds[15–25].

Here, we developed a high-throughput screening (HTS) system to find small-molecular-weight compounds that would function as SOD1$^{mut}$-Derlin-1 interaction inhibitors. By utilizing time-resolved fluorescence resonance energy transfer (TR-FRET) technology, we successfully established a high-throughput, robust assay system to measure the SOD1-Derlin-1 interaction. We performed an HTS of 160,000 compounds in the public chemical library (at the Drug Discovery Initiative (DDI), The University of Tokyo) and found that an analog of one of the hit compounds prevent 122 types of SOD1$^{mut}$ from interacting with Derlin-1. The inhibitor significantly ameliorates the ALS pathology in motoneurons derived from patient induced pluripotent stem cells (iPSCs) and delays the onset and prolongs the survival (14.5% and 14.2% improvement, respectively) of ALS model mice expressing SOD1$^{G93A}$. Our data emphasize the importance of the SOD1-Derlin-1 interaction as a common mechanism of motoneuron toxicity in SOD1$^{mut}$, and we provided a potential mechanism-based ALS treatment.

## Results

**TR-FRET-based HTS assay for the PPI inhibitors.** To identify the inhibitors of the SOD1-Derlin-1 interaction, we first established an HTS system. We employed a TR-FRET-based assay for the detection of SOD1-Derlin-1 interaction by using antibodies labeled with either a donor or an acceptor fluorophore (Fig. 1a). Theoretically, when SOD1 and Derlin-1 are interacting, a donor fluorophore (Europium cryptate: Eu) and an acceptor fluorophore (d2) are in close proximity and generate the FRET signal. In contrast, there is no FRET signal when the small molecules inhibit the SOD1-Derlin-1 interaction. We previously mapped each binding site: SOD1 (5-18), the amino-terminal 14 amino acids of SOD1$^{WT}$, was identified as the minimum essential DBR, and the cytosolic carboxyl-terminal 12 amino acids, Derlin-1 (CT4), specifically interacts with SOD1$^{mut}$ [13,14]. To identify the combinations that efficiently generate FRET signal, lysates from HEK293A cells transfected with various combinations of SOD1 and Derlin-1 constructs including each binding site were mixed with the antibodies labeled with either Eu or d2. Among the over 1000 combinations of constructs and antibodies tested (Supplementary Table 1), we obtained a robust signal from the cell lysate expressing Flag-SOD1$^{G93A}$ and Derlin-1-HA using an anti-Flag antibody labeled with Eu (Flag-Eu) and an anti-HA antibody labeled with d2 (HA-d2), which was sufficient for compound screening (Z′-factor >0.9) (Fig. 1b). To confirm that the FRET signal was derived from the interaction of SOD1$^{G93A}$ with Derlin-1, we performed the competition assay using each binding-site peptide. Both the DBR-containing peptide (SOD1 (5-20); amino-terminal 16 amino acids of SOD1) and the Derlin-1 (CT4) peptide attenuated the FRET signal in a concentration-dependent manner, clearly showing the requirement of the SOD1-Derlin-1 interaction for generating the FRET signal (Fig. 1c).

We screened a total of approximately 160,000 small molecules using the inhibition of FRET signal as a readout. The Z′-factor of all plates in the first screening was set to be over 0.75, suggesting that this assay system was sufficiently robust and reliable (Supplementary Figure 1a). In this stage, 1460 compounds that showed over 15% inhibition were selected for further validation in the second screening (Fig. 1d and Supplementary Figure 1b). The decrease in FRET signal may occur not only because of the inhibition of SOD1-Derlin-1 interaction but also because of the inhibition of interactions between the tags and the corresponding antibodies. Alternatively, compounds may also target the energy transfer itself by interacting with the emission or excitation processes. To exclude such false-positive compounds, we prepared the monomolecular FRET assay using HA-SOD1$^{WT}$-Flag with Flag-Eu and HA-d2 as a counter assay (Supplementary Figure 1c). In the second screening, we reassessed the inhibition of the SOD1-Derlin-1-derived FRET signal of 1460 positive compounds in the first screening and simultaneously checked the effect of each compound on the monomolecular FRET signal of HA-SOD1$^{WT}$-Flag. Many of the compounds that were reconfirmed to inhibit the SOD1-Derlin-1-derived FRET signal also inhibited the monomolecular FRET signal (Fig. 1e, lower panel, region A). However, some compounds attenuated only the SOD1-Derlin-1-derived FRET signal with little effect on monomolecular FRET, suggesting that these compounds inhibit the SOD1-Derlin-1 interaction (Fig. 1e, lower panel, region B). Therefore, 131 compounds were selected, each showing over 15% inhibition of the SOD1-Derlin-1-derived FRET signal without inhibiting the monomolecular FRET signal by more than 15% (Fig. 1e, lower panel, region B and Supplementary Figure 1b).

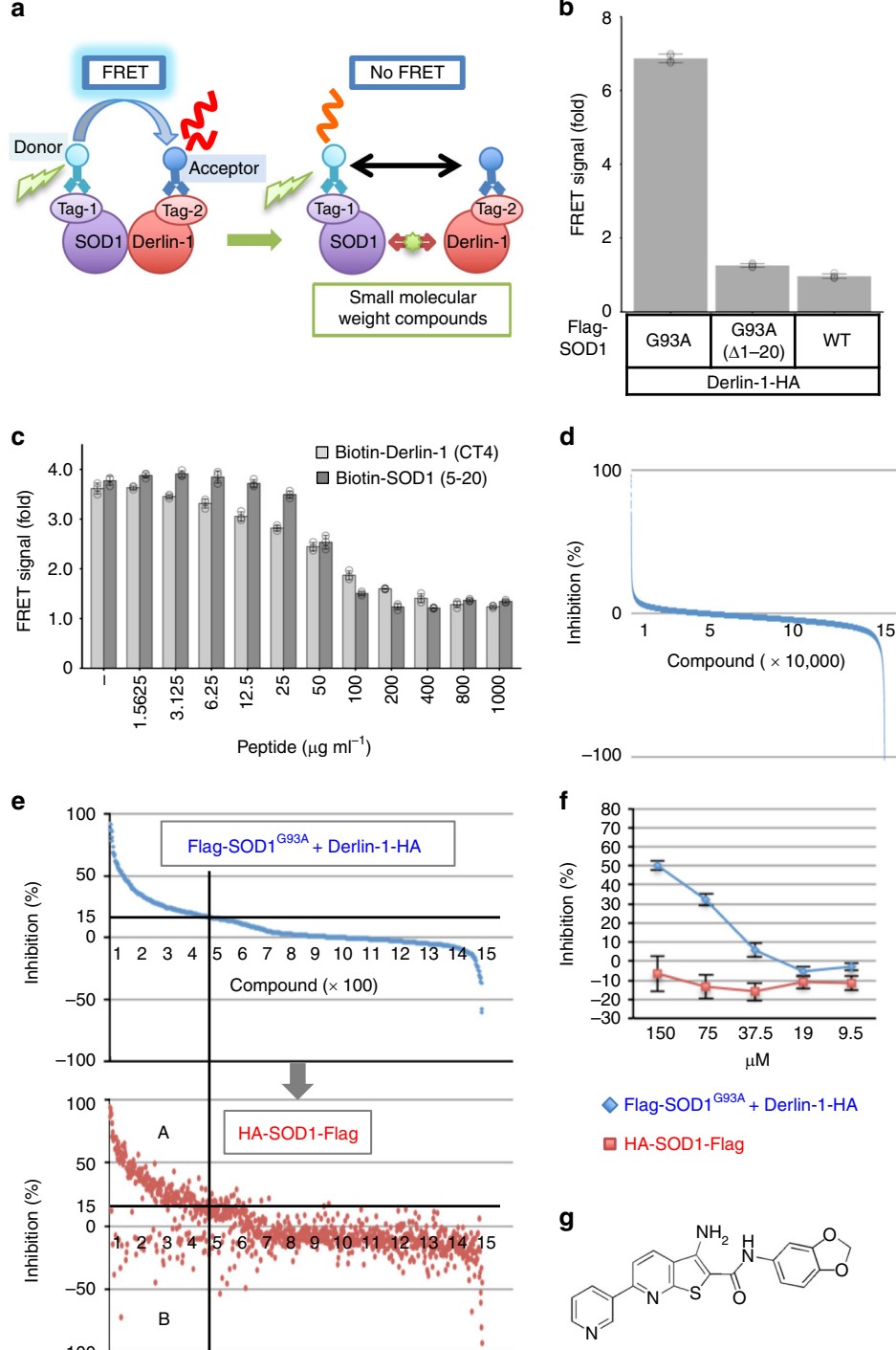

**Fig. 1** Screen of small molecules for SOD1-Derlin-1 interaction inhibitors. **a** TR-FRET-based interaction assay model showing the case of tagged SOD1 as a target of the Eu-labeled antibody and tagged Derlin-1 as a target of the d2-labeled antibody. **b**, **c** FRET signals with Flag-Eu and HA-d2 shown as fold changes from the FRET signal in non-transfected lysate. The data are shown as the mean ± s.d. **b** FRET signal using lysates from HEK293A cells transfected with plasmids as indicated ($n = 4$). **c** Competition assay of FRET signal generated by Flag-SOD1$^{G93A}$ and Derlin-1-HA using indicated concentration of SOD1 (5-20) and Derlin-1 (CT4) peptide. Lysates from HEK293A cells transfected with Flag-SOD1$^{G93A}$ and Derlin-1-HA were incubated with the indicated concentration of each peptide for 16 h ($n = 3$). Light gray: with Derlin-1 (CT4) peptide; dark gray: with SOD1 (5-20) peptide. **d** Result of the first screening. The compounds are presented in order of inhibition rate. **e**, **f** Blue: inhibition (%) against FRET signal generated by Flag-SOD1$^{G93A}$ and Derlin-1-HA; red: inhibition (%) against FRET signal generated by HA-SOD1$^{WT}$-Flag. **e** Result of the second screening. The compounds are presented in order of inhibition rate against the SOD1-Derlin-1-derived FRET signal. **f** Result of #56 in the third screening. The data are shown as the mean ± s.d. ($n = 4$). **g** Chemical structure of #56

Using a titration assay as the third screening, 44 compounds that inhibited the SOD1-Derlin-1-derived FRET signal in a concentration-dependent manner were further selected as positive compounds (Supplementary Figure 1b, d).

**In vitro compound validation identified #56 analogs**. In the FRET assay, positive compounds may still include compounds that alter only the conformation of SOD1 or Derlin-1 and thereby disrupt FRET signal without dissociation between the two proteins. Thus, we performed an in vitro co-immunoprecipitation assay by adding the compounds to the purified Flag-SOD1[G93A]-Derlin-1-HA complex. Twelve compounds out of 44 became positive in this assay (Supplementary Figure 2a–e). Among them, we focused on one of the most prominent inhibitors, named compound #56, because of its dose-dependency in the FRET screening and its drug-like chemical structure defined according to the Lipinski's rule of five, the absence of reactive functional groups, and the exclusion of promiscuous hitters[26,27]. The easiness to speculate the pharmacophore and to synthesize the analogs was also taken into consideration (Fig. 1f, g and Supplementary Figure 2d).

Structural analogs of compound #56 were commercially available. More than 200 of those compounds were examined for potential inhibitory effects on SOD1-Derlin-1 interaction. Some of the #56 analogs that showed over 15% inhibition of the SOD1-Derlin-1-derived FRET signal also inhibited the interaction between SOD1 and Derlin-1 in in vitro immunoprecipitation assay (Fig. 2a). Moreover, we found inhibitors that showed lower half-maximal inhibitory concentration (IC$_{50}$) values than #56 itself (Fig. 2b), including #56-20 and #56-26 (Figs. 2c, d, respectively). These data suggest that the basic core structure of #56 analogs has an inhibitory effect on the SOD1-Derlin-1 interaction.

**#56-59 disrupts all SOD1$^{mut}$-Derlin-1 interactions analyzed**. Unfortunately, none of the #56 analogs tested above showed inhibitory activities in the cell-based co-immunoprecipitation assay. To evaluate the possibility that the undesired interaction with serum-derived substances was the cause of the inactivation, we assessed the activity in serum-starved condition. Within the serum-depleted medium, some of the compounds showed the inhibition activity, suggesting that the undesired interaction in the culture medium was one of the causes of the inactivity (Supplementary Figure 3a). In addition, #56-26 was relatively stable and showed low permeability, suggesting that the inability might also be due to the low permeability of the plasma membrane (Supplementary Figure 3b).

Although there were some compounds that inhibited the SOD1-Derlin-1 interaction in serum-depleted condition, we have previously reported that the serum depletion causes the conformational change in SOD1[14,28]. Thus, we synthesized new #56 derivatives to obtain novel inhibitors that are active in the

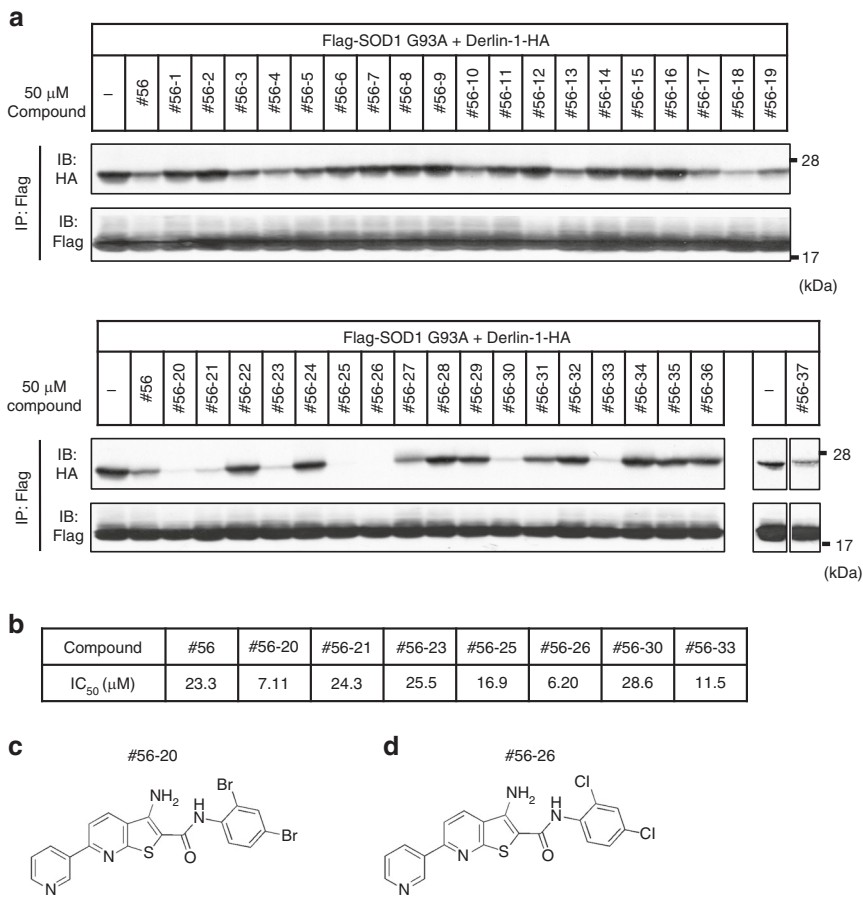

**Fig. 2** Compound validation via in vitro assays. **a** Inhibition of SOD1$^{G93A}$-Derlin-1 interaction by #56 analogs in the in vitro immunoprecipitation (IP) assay. Purified SOD1$^{G93A}$-Derlin-1 complex form HEK293A cell lysates transfected with Flag-SOD1$^{G93A}$ and Derlin-1-HA were incubated with the indicated compounds for 16 h. Then, the complex was immunoprecipitated using anti-Flag beads, followed by immunoblotting (IB) analysis with the indicated antibodies. **b** IC$_{50}$ values of notable #56 analogs and #56 in the in vitro assay. IC$_{50}$ values were calculated by generating standard curves. **c, d** Chemical structure of #56-20 and #56-26

culture medium containing serum. In an in vitro immunoprecipitation assay, two (#56-40, #56-41) out of six new compounds worked as well as #56 itself, correlating with the result of FRET-based assay (Fig. 3a, b and Supplementary Figure 3c). Moreover, #56-40 showed an improved permeability and inhibitory effect on

the SOD1-Derlin-1 interaction even in a cell-based co-immunoprecipitation assay (Fig. 3c and Supplementary Figure 3b). However, this compound has the inconsistent effect, suggesting that the effective concentration depends on cellular conditions. Therefore, we synthesized additional compounds and found a

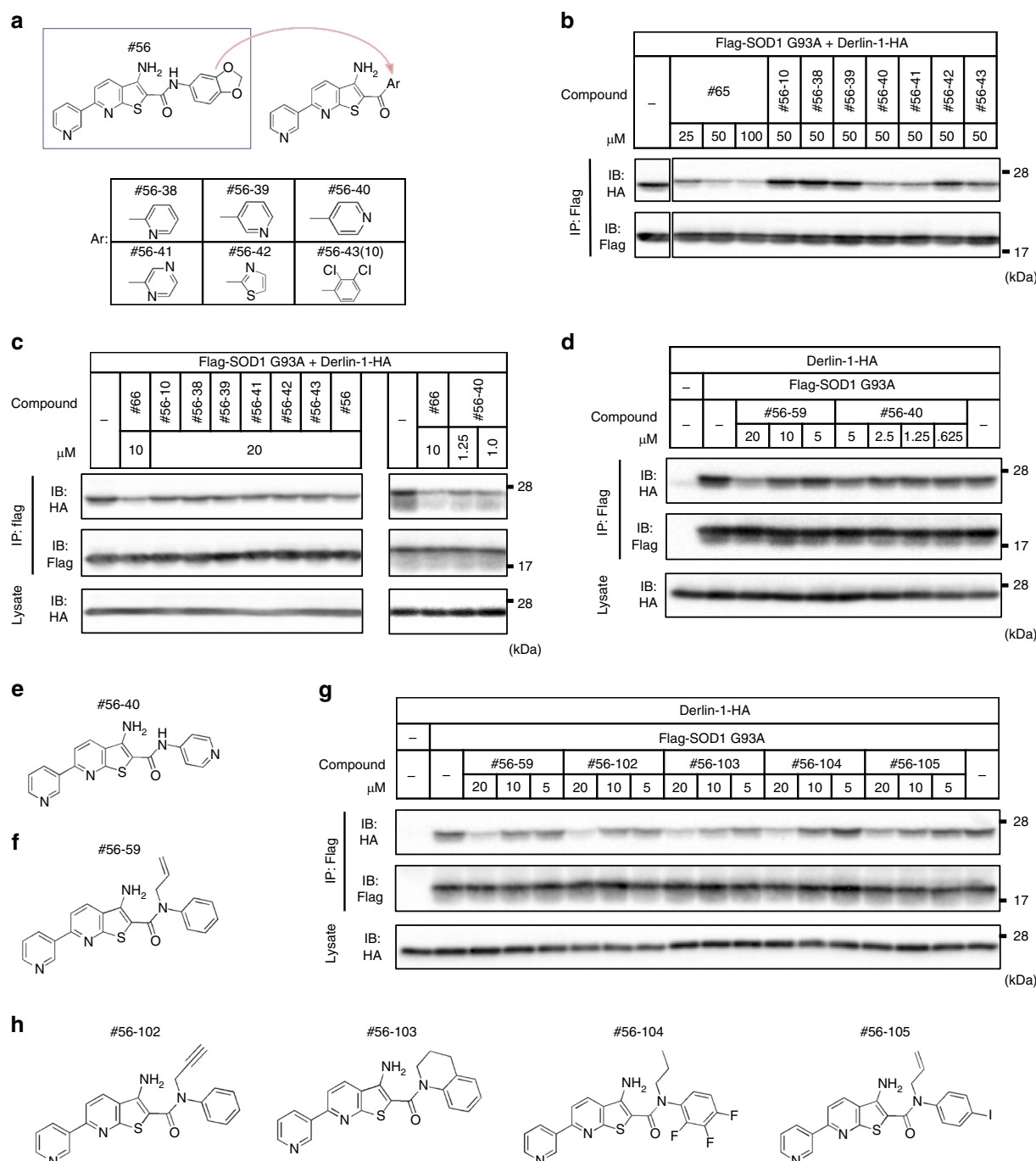

**Fig. 3** Identification of cell-permeable compounds. **a** Chemical structure of newly synthesized #56 analogs. #56-43 is the resynthesized #56-10. **b** Inhibition of SOD1$^{G93A}$-Derlin-1 interaction by new #56 analogs in in vitro IP assay. The SOD1$^{G93A}$-Derlin-1 complex purified from lysates transfected with Flag-SOD1$^{G93A}$ and Derlin-1-HA was incubated with the indicated compounds for 16 h and analyzed by IP-IB with the indicated antibodies. **c**, **d** Inhibition of SOD1$^{G93A}$-Derlin-1 interaction by #56 analogs in a cell-based IP assay. HEK293A cells transfected with indicated plasmids were treated with the indicated compounds for 24 h, and lysates were analyzed by IP-IB with the indicated antibodies. **c** Inhibition of SOD1$^{G93A}$-Derlin-1 interaction by the same compounds as in the in vitro assay **b** in a cell-based assay. Compound #66 is used as a positive control of a cell-permeable inhibitor, although the resynthesized #66 does not show any activities. **d** Two cell-permeable SOD1$^{G93A}$-Derlin-1 interaction inhibitors in a cell-based IP assay. **e**, **f** Chemical structure of #56-40 and #56-59. **g** Inhibitory activities of other compounds containing tertiary amide groups, #56-102-105, in the cell-based IP assay as in **c**, **d**. **h** Chemical structure of #56-102, 103, 104, and 105

more stable inhibitor in cell-based assay, #56-59 (Fig. 3d–f). Through the further evaluation of newly synthesized #56 derivatives, we found that the character of potent inhibitors in cell-based assay is provided by the replacement of hydrogen with alkyl groups at the amide bond, turning that amide into a tertiary amide (Fig. 3g, h). Testing several derivatives with a tertiary amide, we found even more potent inhibitor, #56-111 (Supplementary Figure 4a, b).

SOD1$^{G93A}$, which we used for the above screening steps, is a representative SOD1$^{mut}$ among 122 types of ALS-related SOD1$^{mut}$ that interact with Derlin-1. Thus, we next examined whether #56-59 could also inhibit the interactions of Derlin-1 with 122 types of SOD1$^{mut}$ in the cell-based co-immunoprecipitation assay. #56-59 clearly attenuated the interactions between Derlin-1 and all tested SOD1$^{mut}$ (Supplementary Figure 5a–h). These results suggest that ALS-related SOD1$^{mut}$ commonly possess a similar mechanism of interaction with Derlin-1, consistent with our previous findings of the importance of DBR exposure in SOD1 for the interaction with Derlin-1[14,28,29]. SOD1$^{WT}$ is known to form homodimers, and Derlin-1 forms homomeric and heteromeric complexes with Derlin family proteins and with other ERAD components such as Hrd1[30]. Under the same conditions as those in which #56-59 inhibited the SOD1-Derlin-1 interaction, these interactions were not affected, suggesting that #56-59 inhibits the SOD1-Derlin-1 interaction with some specificity (Supplementary Figure 6).

**#56 analogs target SOD1 DBR**. There are two putative modes of action of PPI inhibitors: intervention by direct binding to the interface or allosteric interference. To reveal the mechanism of #56-mediated PPI inhibition, we evaluated the inhibition activity of #56 analogs against the SOD1-Derlin-1 interaction using full-length proteins and/or fragments containing binding sites in the in vitro assay. Due to the low activity of #56-59 in vitro, two prominent in vitro inhibitors, #56-20 and #56-26, were assessed. Both of them inhibited all combinations of the interaction (Fig. 4a). The inhibition of the interaction between SOD1 (1-20) and Derlin-1 (CT4) suggested that #56 analogs exert their activities by interacting with the binding region of either SOD1 or Derlin-1. To identify the direct binding target of #56 analogs, we tried the surface plasmon resonance (SPR) method with biotinylated peptide as a ligand and #56 analogs as an analyte. However, the interaction could not be detected probably due to the relatively weak interaction and low solubility of the compounds (Supplementary Figure 7a). Thus, we next performed fluorescence polarization (FP) assays in the presence of SOD1 (5-20) peptide or Derlin-1 (CT4) peptide, utilizing the fluorescence properties of #56-20 per se. The degree of FP increased in the presence of SOD1 (5-20) peptide rather than the Derlin-1 (CT4) peptide, suggesting that #56-20 preferentially and directly interacts with the SOD1 DBR (Fig. 4b). We also compared the binding ability of #56-20 to the full-length recombinant SOD1$^{WT}$ and SOD1$^{G93A}$. The preferential interaction of #56-20 with SOD1$^{G93A}$ was observed ($K_d = 81.1 \mu M$ and Supplementary Figure 7b). This is consistent with our model in which SOD1$^{mut}$ possess a DBR-exposed conformation and that #56-20 interacts with the DBR[14,29].

Next, to confirm the target of the compounds within the cells, we embedded benzophenone as a photoactivatable crosslinking moiety, and a biotin as a tag to #56-59, a potent inhibitor in cell-based assay. We named this compound Photo-Biotin-PEG$_3$-#56-59, PB56 for short (Fig. 4c, right). Although it was less potent than the original compound (#56-59), PB56 also inhibited the SOD1-Derlin-1 interaction in a dose-dependent manner (Supplementary Figure 7c). In the crosslinking and pull-down assays

using SOD1$^{G93A}$-transfected or Derlin-1-transfected cells treated with PB56, we clearly observed the biotin-labeled band at the same position of monomer Flag-SOD1$^{G93A}$ and high-molecular smear bands probably representing sodium dodecyl sulfate (SDS)-resistant SOD1$^{G93A}$ aggregates, suggesting that PB56 selectively binds to SOD1 and not Derlin-1 (Fig. 4c, left). Pre-treatment with #56-59 competitively displaced the binding of PB56 to SOD1$^{G93A}$, indicating that the common basic structure of #56-59 in PB56 mediated the interaction with SOD1 (Fig. 4d). Moreover, we confirmed that PB56 interacted with SOD1$^{G93A}$ more strongly than SOD1$^{WT}$ within the cells (Fig. 4e).

Additionally, we evaluated the effects of #56 analogs to SOD activity in vitro. Both #56-20 and #56-26, which possess strong inhibitory activity on the SOD1-Derlin-1 interaction in vitro, showed no effects on SOD activity (Supplementary Figure 7d). Taken together, these data suggest that #56 analogs might specifically inhibit the interaction with Derlin-1 by directly binding to the SOD1 DBR without affecting the enzymatic activity of SOD1.

**#56-59 ameliorates the pathology of ALS**. Finally, we evaluated the therapeutic effects of #56 analogs in in vitro human model utilizing ALS patient iPSCs as well as in in vivo ALS mouse model. Recently, we have characterized and established the in vitro ALS model system by utilizing patient iPSCs[31]. Therefore, we treated #56-40 and #56-59 to motoneurons derived from ALS patient iPSCs with L144FVX mutation in SOD1 (ALS1). Although #56-40 was less effective, probably because of the narrow effective dose, treatment with #56-40 and #56-59 increased the survival of ALS motoneurons (Fig. 5a). Then, we further investigated the effect of #56-59 to other iPSC-derived motoneurons. The number of motoneurons generated from healthy control-derived iPSC was not increased in the presence of #56-59, indicating that the increase of ALS motoneurons with #56-59 was not due to the improvement of differentiation efficiency or survival independent of ALS pathology (Supplementary Figure 8a). Although #56-59 did not improve the viability of motoneurons generated from ALS3 (SOD1$^{G93S}$), it restored the motoneuron survival of ALS2 (SOD1$^{L144FVX}$) (Supplementary Figure 8a). These data suggested that #56-59 could ameliorate SOD1$^{mut}$ toxicity in in vitro human ALS model. The conformational changes in SOD1$^{WT}$ have been reported in FALS with other ALS-causative gene mutations, including FUS and TDP-43 [32]. Thus, we examined the effect of #56-59 on motoneuron toxicity with TDP-43 mutant; however, #56-59 failed to improve the survival of ALS4 motoneurons (TDP-43$^{M337V}$) at least in this condition (Supplementary Figure 8a).

Next, we performed continuous delivery of the compounds to SOD1$^{G93A}$ transgenic male mice by using osmotic pumps. Because we were concerned that the efficacy of the compounds might be compromised by a limited ability to cross the blood–brain barrier and to access the target motoneurons, intracerebroventricular (i.c.v.) cannulation was chosen as the method for delivery. The start point of administration was set at 22 weeks of age, approximately 6 weeks before the usual onset timing as defined in our previous study[13]. We infused the mice with dimethyl sulfoxide (DMSO) as control, 1 mM #56-40, and 3 mM #56-59 at a flow rate of 0.15 $\mu$l h$^{-1}$. The onset, defined as motor function deficit observed in rotarod performance, and the survival time were monitored. While mice infused with #56-40, which might be out of effective dose, were comparable to control mice, mice infused with #56-59 showed significantly delayed onset, by a median of 4.5 weeks (14.5% improvement), and also showed significantly prolonged survival, by a median of 5 weeks (14.2% improvement) (Fig. 5b, Supplementary Figure 8b, and

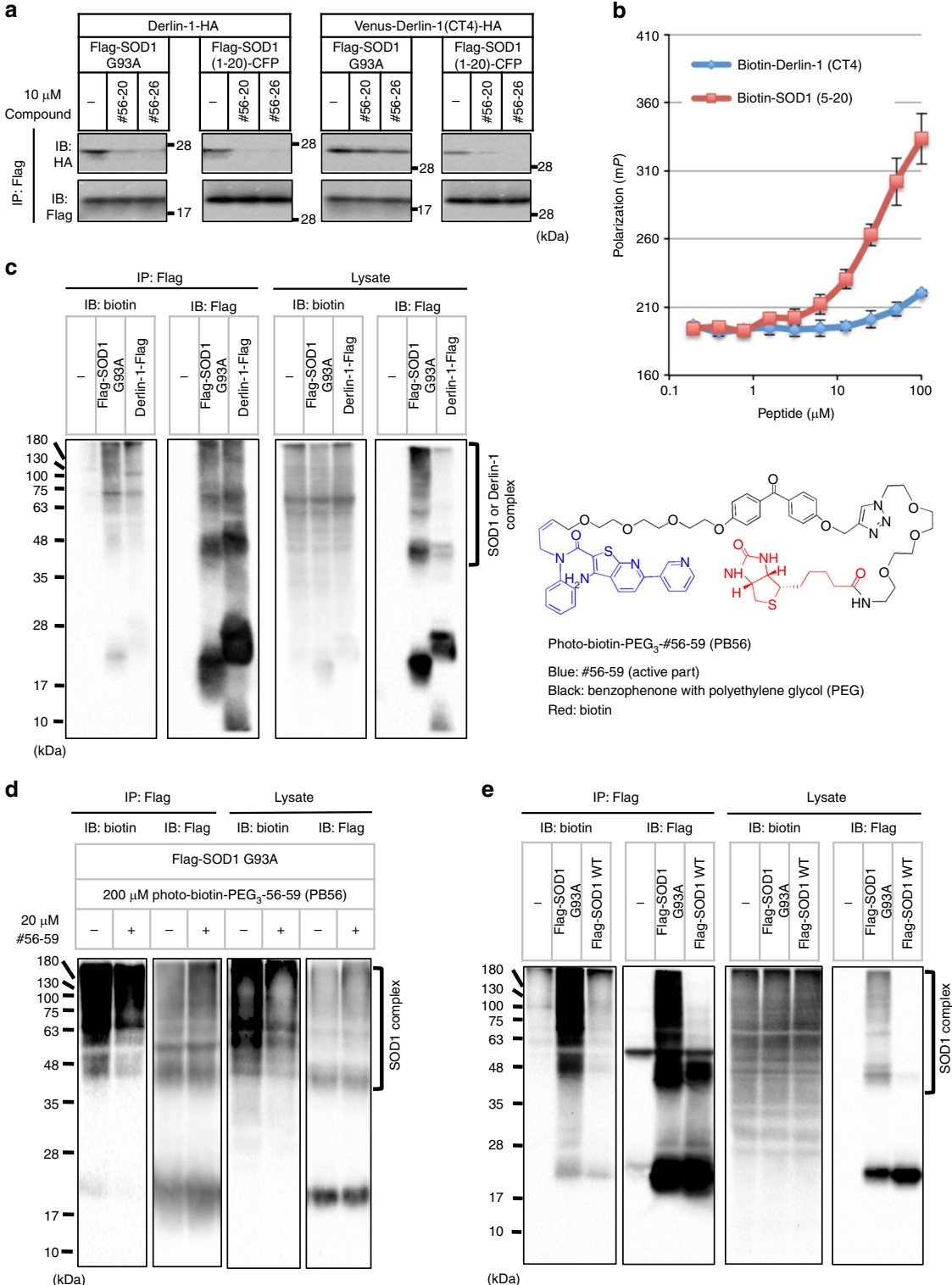

**Fig. 4** #56 analogs target SOD1 DBR. **a** In vitro IP assay with various SOD1-Derlin-1 complexes. SOD1-Derlin-1 complexes purified from lysates transfected with the indicated plasmids were incubated with #56-20 or #56-26 for 16 h and analyzed by IP-IB with the indicated antibodies. **b** FP analysis of #56-20 in the presence of indicated concentration of SOD1 (5-20) and Derlin-1 (CT4) peptide. Here, 250 nM #56-20 was incubated with the indicated concentration of each peptide for 6 h, and FP was measured. Blue: Derlin-1 (CT4); red: SOD1 (5-20). The data are shown as the mean ± s.d. ($n = 3$). **c–e** Pull-down assay using PB56. HEK293A cells transfected with indicated plasmid were treated with 200 μM PB56 for 24 h followed by UV light irradiation for 1 h. Lysates were analyzed by IP-IB with the indicated antibodies. **c** Pull-down assay by Flag-SOD1[G93A] or Derlin-1-Flag. Chemical structure shows PB56 (blue: #56-59 (active part); black: benzophenone with polyethylene glycol (PEG) as spacers; red: biotin). **d** Competition assay with the pre-treatment of 20 μM #56-59 for 12 h. **e** Pull-down assay by Flag-SOD1[G93A] or Flag-SOD1[WT]

Supplementary Table 2). Consistent with these results, the number of motoneurons detected by Nissl staining of lumbar spinal cord sections at 31 weeks of age was significantly increased in #56-59-treated ALS model mice (Fig. 5c, d). These data clearly show that the SOD1-Derlin-1 interaction inhibitor can ameliorate ALS pathology both in in vitro human model and in vivo mouse model, demonstrating the importance of the SOD1-Derlin-1 interaction in the pathogenesis of SOD1[mut]-induced FALS and the potential of the SOD1-Derlin-1 interaction as a therapeutic target in ALS.

## Discussion

In the present study, we designed and developed a high-throughput, robust screening assay system for measuring the

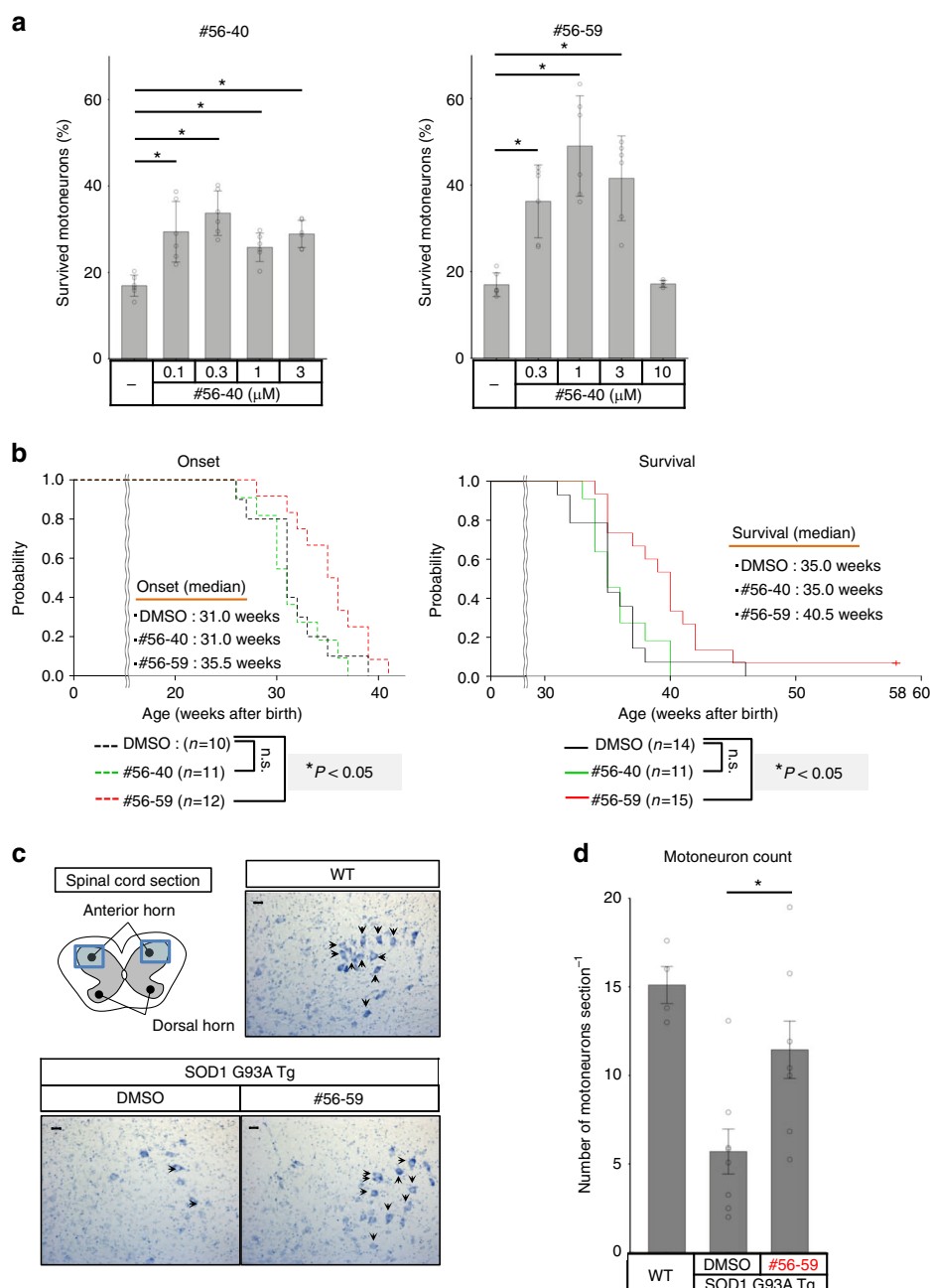

**Fig. 5** Inhibiter of SOD1-Derlin-1 interaction, #56-59, ameliorates ALS pathology. **a** Effects of #56 analogs on motoneurons derived from ALS patient iPSCs with L144FVX mutation in *SOD1* (ALS1). The ratios of surviving motoneurons (day 14/day 7 (%)) are shown as the mean ± s.d. (n = 6; *P < 0.05, also see Methods for statistical test). **b** Analysis of onset (DMSO: n = 10, #56-40: n = 11, #56-59: n = 12) and survival (DMSO: n = 14, #56-40: n = 11, and #56-59: n = 15) in SOD1[G93A] transgenic mice (SOD1 G93A Tg) continuously infused with #56 analogs by i.c.v. cannulation. Onset was determined by motor function deficit seen in rotarod performance. Black: DMSO treatment (control); green: #56-40 treatment; red: #56-59 treatment. Mice were followed in this survival study for 58 weeks, and the survival time of one mouse that did not show paralysis is shown as a tick on the line. Statistical analysis was performed using the Kaplan–Meier method followed by the log-rank test and the Gehan–Breslow–Wilcoxon test (*P < 0.05, n.s.: not significant). **c** Representative Nissl stain image of the lumbar spinal cord section (L2–L5) of indicated mice. All motoneuron counts were performed in a blinded fashion. Scale bar = 20 μm. **d** Quantification of motoneuron counts in **c**. Five sections per mouse were counted, and the data are shown as the mean ± s.e.m. WT mice: n = 4; DMSO-treated and #56-59-treated SOD1 G93A Tg: n = 8 in each group, *P < 0.05: significance was calculated using Student's t test

interaction between two proteins, SOD1 and Derlin-1 (Fig. 1a–c). We screened approximately 160,000 compounds and selected one potential scaffold, #56 (Fig. 1d–g). We found that some analogs of #56 also possessed inhibitory activities in vitro (Fig. 2a–c). Moreover, newly synthesized #56 analogs inhibited the SOD1-Derlin-1 interaction in cell-based assays (Fig. 3c, d, g). One of these inhibitors, #56-59, exerted its activity on all types of $SOD1^{mut}$-Derlin-1 interaction that we previously reported[14] (Supplementary Figure 5a–h). Furthermore, we show that the SOD1-Derlin-1 interaction inhibitor can ameliorate ALS pathology both in in vitro human model and in vivo mouse model (Fig. 5).

We used two inhibitors, #56-40 and #56-59, to assess the effect to the ALS pathology. However, unlike #56-59, #56-40 showed only modest effects to the ALS pathology (Fig. 5a, b). Our concern was that the effective concentration of #56-40 was in a very narrow range. Thus, we assume that the dose of #56-40 might be insufficient to show a therapeutic effect on ALS model mice under these conditions. Moreover, ALS4 motoneurons showed even a vulnerability to #56-59. The ALS motoneurons would be feeble, and the effective dose of #56-59 might be different among iPSC lines. The failure of improvement in ALS3 and the significant reduction in ALS4 could be caused by the toxicity of #56-59.

The detected concentration of #56-59 in the brain and spinal cord of the mice were very low (Supplementary Figure 8b and Supplementary Table 2). In addition, we could not evaluate the inhibition activity in vivo, because the level of the SOD1-Derlin-1 interaction varies even in the non-treated ALS model mice (Supplementary Figure 9a). Taking account of the unbound fractions within the serum, the effective concentration of #56-59 could be estimated between 0.25 to 1 μM in cell-based immunoprecipitation assay (Supplementary Figure 4a and Supplementary Figure 8c). One of the possible reasons of this discrepancy is the modification of #56-59. #56-40 showed a relatively unstable character (Supplementary Figure 3b). Although HEK293A cells and the cells in central nervous system do not express typical metabolizing enzymes compared to hepatic cells, #56-40 and probably #56-59 could be metabolized in cells and in vivo. In the present method that we measured the concentration of #56-59, the modified compounds could not be estimated. Therefore, it is possible that the modified #56-59, which we could not identify in this study, contributed to the inhibition of SOD1-Derlin-1 interaction in the present assays.

Another possibility is the difference of the treatment time course. In the long-term treatment in vivo, #56-59 could occupy the DBR of newly synthesized $SOD1^{mut}$ before its interaction with Derlin-1. Thus, #56-59 would be able to efficiently inhibit the SOD1-Derlin-1 interaction compared to the cell-based immunoprecipitation assay, where #56-59 might dissociate the pre-formed complexes. It is also possible that as we could find the increase of #56-59 in spinal cord at 5 weeks of infusion (Supplementary Figure 8b and Supplementary Table 2), the longer continuous infusion would reach higher concentration.

Notably, one ALS model mouse treated with #56-59 did not show any symptoms until 58 weeks of age. Although it is unclear whether #56-59 could really suppress the pathology completely in some population or whether additional factor(s) might exist, further improvement of the administration method, including doses, timing of initial administration, and pharmacokinetics, should provide us with more effective treatment for ALS pathologies.

In the present study, we administered the compounds before the appearance of defects in motor function. Our compound delayed the onset of the disease, suggesting that the SOD1-Derlin-1 interaction is involved in the onset of ALS, and earlier treatment might improve the phenotype more effectively. The early

diagnosis of conformation-disordered SOD1 using our ELISA system that can easily detect the DBR-exposed SOD1 may enable us to perform a differential diagnosis of the applicable patients before onset and to provide pre-onset treatment[14,26].

We have previously reported that ER stress evoked by the SOD1-Derlin-1 interaction activates ASK1, and ASK1 deficiency prolonged the survival of ALS model mice by improving the disease progression but not onset[13]. Moreover, the administration of ASK1 inhibitors showed the reduction of the glial cell activity[33], which is known to contribute to the progression of the pathology[34,35]. On the other hand, the #56-59 treatment had little effect on the progression and did not show significant suppression of gliosis (Supplementary Figure 9b). Thus, we estimate that the SOD1-Derlin-1 interaction may primarily target the signaling pathways that contribute to the disease onset independent of ASK1 pathway. It would be an important issue to reveal the detailed signaling pathways affected in the present treatment protocol and the role of ASK1 in non-cell autonomous motoneuron death, as well as the effects of #56-59 on the ASK1 pathway in in vitro and in vivo.

#56 analogs preferentially interact with $SOD1^{mut}$ rather than $SOD1^{WT}$. Although we cannot exclude the possibility that #56 analogs inhibit the interaction of $SOD1^{mut}$ through the DBR not only with Derlin-1 but also with other proteins, the SOD1-Derlin-1 interaction is the most likely and promising target for ALS treatment. It is also important to carefully interpret the specificity of #56 analogs in cells or in vivo because of the very weak $K_d$ value of them (Supplementary Figure 7a). Further improvements of #56 analogs' affinity to $SOD1^{mut}$ would be required. The elucidation of compound-SOD1 complex atomic structure should resolve these questions and facilitate further optimization of the compounds.

We also found that zinc deficiency caused a conformational change of $SOD1^{WT}$ to a mutant-like form (exposure of the DBR), resulting in SOD1-Derlin-1 interaction[28]. Recently, it was reported that conformationally disordered $SOD1^{WT}$ was observed in SOD1 mutation-negative sporadic ALS (SALS) patients[36]. $SOD1^{WT}$ has also been shown to be required for non-cell autonomous motoneuron toxicity in astrocytes or oligodendrocytes derived from SALS patients[37,38]. These data suggest that under certain conditions, including genetic factors and environmental factors, $SOD1^{WT}$ may also serve as the cause of ALS pathogenesis and that inhibition of the SOD1-Derlin-1 interaction may be a potential target for ALS therapy development not only for $SOD1^{mut}$-caused FALS but also for a subset of SOD1 mutation-negative FALS and SALS, in which conformationally disordered $SOD1^{WT}$ is proposed to be involved. To approach this possibility, we investigated the effect of #56-59 on the iPSC-derived motoneurons with TDP-43 mutant. However, consistent with our previous observation that the exogenously expressed TDP-43 mutants could not alter the $SOD1^{WT}$ conformation[29], administration of #56-59 did not improve motoneuron viability (Supplementary Figure 8a). Since we cannot exclude the possibility that the vulnerability masked the positive effect of #56-59 in ALS4, it is important to evaluate the involvement of the SOD1-Derlin-1 interaction by using multiple inhibitors, different iPSC-derived motoneurons, and/or other methods.

In summary, we succeeded in providing a proof-of-concept regarding our SOD1-Derlin-1 interaction hypothesis by ameliorating the disease both in in vitro human model and in vivo mouse model by using a developed PPI inhibitor. These data re-emphasize the importance of the SOD1-Derlin-1 interaction as a common mechanism of motoneuron toxicity in $SOD1^{mut}$, and we provided a potential approach for a molecular mechanism-based ALS treatment.

## Methods

**Plasmids.** The complementary DNAs (cDNAs) encoding plasmids used in developing the TR-FRET-based binding assay (Supplementary Table 1a, b) were constructed in the pcDNA3.0 plasmid (Invitrogen) by PCR. The pcDNA3.0-Derlin-1-HA, pcDNA3.0-Venus-Derlin-1(CT4)-HA, pcDNA3.0-Derlin-1-Flag, pcDNA3.0-Derlin-2-Flag, pcDNA3.0-Derlin-3-Flag, pcDNA3.0-Flag-HRD1, pcDNA3.0-Flag-SOD1$^{WT}$, and pcDNA3.0-Flag-SOD1$^{mut}$, and pcDNA3.0-Flag-SOD1 (1-20)-CFP plasmids have been constructed in previous studies[13,14]. Transfection was performed using Polyethylenimine-Max (Polysciences) according to the manufacturer's instructions.

**Antibodies and peptides.** Anti-tag antibodies labeled with either Eu or d2 (Supplementary Table 1c) were purchased from Cisbio. Horse radish peroxidase (HRP)-linked anti-biotin antibody (7075, Cell Signaling Technology, dilution 1:2000) and antibodies against Flag tag (F3165, Sigma, dilution 1:10,000), HA tag (11867431001, Roche, dilution 1:10,000), βIII-tubulin (MRB-435P, Covance, dilution 1:2000), GFAP (N1506, DAKO, dilution 1:10), and Iba1 (019-19741, Wako, dilution 1:500), SOD1 (ADI-SOD-100, Enzo Life Science, dilution 1:5000), and Derlin-1 (SAB4200148, Sigma, 1.5 µg for immunoprecipitation) were purchased from the indicated suppliers. The Derlin-1 antibody has been generated previously (dilution 200 ng ml$^{-1}$)[13]. The biotinylated SOD1 (5-20) peptide (biotin-VCVLKGDGPVQGIINF) and biotinylated Derlin-1 (CT4) peptide (biotin-CFLYRWLPSRRGG) were purchased from Eurofins.

**Cell cultures.** HEK293A cells were purchased from Invitrogen and were used due to the high efficiency of lipofection. The cells were cultured in Dulbecco's modified Eagle's medium (DMEM) containing 10% fetal bovine serum, 4.5 mg ml$^{-1}$ glucose, and 100 U ml$^{-1}$ penicillin, and maintained in an atmosphere of 5% CO$_2$ at 37 °C. HEK293A cells were tested for the presence of mycoplasma and identified by morphology and efficient production of adenovirus.

**TR-FRET-based interaction assay.** HEK293A cells transfected with Flag-SOD1$^{G93A}$ and Derlin-1-HA or HA-SOD1$^{WT}$-Flag were lysed in FRET buffer containing 25 mM phosphate buffer pH 7.0, 400 mM potassium fluoride, 0.1% bovine serum albumin (BSA), 0.5% Triton X-100, 5 µg ml$^{-1}$ leupeptin, 1 mM phenylmethylsulfonyl fluoride. In competition assay using biotinylated SOD1 (5-20) peptide and Derlin-1 (CT4) peptide, lysates from HEK293A cells transfected with Flag-SOD1$^{G93A}$ and Derlin-1-HA were incubated with the indicated concentration of each peptide for 16 h. Then, the fluorophore-labeled antibodies were mixed, and FRET signals were measured. In the compound screen, lysates were added in the 384-well plates (Greiner) in which compounds were dispensed in advance. In the first and second screening steps, compounds were tested at final concentrations of 50 µM by applying 250 nL of 2 mM stock solution. In a 5-point dose–response experiment in the third screening, compound concentrations were 150, 75, 37.5, 19, and 9.5 µM with 0.75% DMSO. After the incubation at 4 °C for 12–16 h, the anti-Flag antibody labeled with Eu and the anti-HA antibody labeled with d2 diluted in FRET buffer were added and incubated for 1 h at room temperature in 384-well plates. Then, the FRET intensity (337 nm excitation filter/665 nm emission filter) and donor intensity (337 nm excitation filter/620 nm emission filter) were measured using PHERAstar (BMG Labtech), and the FRET ratio was calculated as FRET intensity/donor intensity. FRET signal was the FRET ratio of DMSO-treated transfected cell lysates sample (FRET Ratio Max) divided by that of lysate samples from non-transfected cells (FRET Ratio Min). Inhibition (%) was calculated using the following equation: Inhibition (%) = 100−100 {(FRET Ratio of each tested compound) − (FRET Ratio Min)}/ {(FRET Ratio Max) − (FRET Ratio Min)}. The number of samples represents technical replicates.

**Immunoblotting analysis.** HEK293A cells were lysed on ice in a buffer containing 20 mM Tris-HCl, pH 7.5, 150 mM NaCl, 10 mM EDTA, 1% Triton X-100, 5 µg ml$^{-1}$ leupeptin, and 1 mM phenylmethylsulfonyl fluoride. After centrifugation, cell extracts were resolved by SDS-polyacrylamide gel electrophoresis (SDS-PAGE) and electroblotted onto polyvinylidene difluoride membranes. After blocking with 5% skim milk in TBS-T (50 mM Tris-HCl, pH 8.0, 150 mM NaCl, and 0.05% Tween-20), the membranes were probed with antibodies to HA, Flag, α-tubulin, or biotin (HRP-linked). The proteins were detected using the ECL system. IC$_{50}$ values of #56 analogs in in vitro assay were calculated by generating standard curves, and analyses were performed using a ChemiDoc and Image Lab software (Bio-Rad). The uncropped scans were supplied in Supplementary Figure 10. All immunoblotting analyses were repeated at least three times and the same results were obtained.

**In vitro immunoprecipitation analysis.** HEK293A cell lysates transfected with each plasmid were immunoprecipitated with anti-Flag beads (M2 Affinity Gel, Sigma). The beads were washed with washing buffer 1 (20 mM Tris-HCl, pH 7.5, 500 mM NaCl, 5 mM EGTA, and 1% Triton X-100) twice and a buffer containing 20 mM Tris-HCl, pH 7.5, 150 mM NaCl, 10 mM EDTA, and 1% Triton X-100 once. Then, the SOD1-Derlin-1 complex was eluted by elution buffer (0.1% Triton X-100, 500 mM NaCl, 20 mM Tris-HCl, pH 7.5, 5 mM EGTA, and 100 µg ml$^{-1}$ 1× Flag peptide) and ultrafiltered using Centrifugal Filters (Amicon Ultra 10K,

Millipore) to exclude the Flag peptide. After optimal dilution with dilution buffer (0.01% Triton X-100, 500 mM NaCl, 20 mM Tris-HCl, pH 7.5, 5 mM EGTA), SOD1-Derlin-1 complexes were incubated with compounds and immunoprecipitated with anti-Flag beads (M2 Affinity Gel, Sigma). The remaining SOD1-Derlin-1 complexes were separated by SDS-PAGE and analyzed via immunoblotting with antibodies for HA or Flag.

**Cell-based immunoprecipitation analysis.** HEK293A cells transfected with each plasmid were lysed as for the immunoblotting analysis, and lysates were immunoprecipitated with anti-Flag beads (M2 Affinity Gel, Sigma). The beads were washed twice with washing buffer 1 and then once with washing buffer 2 (20 mM Tris-HCl, pH 7.5, 150 mM NaCl, and 5 mM EGTA), separated by SDS-PAGE, and immunoblotted with antibodies for HA or Flag.

**Immunoprecipitation analysis from mouse spinal cord.** The mouse spinal cords were homogenated on ice in a buffer containing 20 mM Tris-HCl, pH 7.5, 150 mM NaCl, 10 mM EDTA, 1% Triton X-100, 5 µg ml$^{-1}$ leupeptin, and 1 mM phenylmethylsulfonyl fluoride. The lysates were immunoprecipitated with anti-Derlin-1 antibody with protein G-Sepharose (GE Healthcare). The beads were washed three times with the lysis buffer, separated by SDS-PAGE, and immunoblotted with antibodies for SOD1 or Derlin-1.

**Compound treatment for immunoprecipitation assay.** For the in vitro immunoprecipitation, compounds were added to purified SOD1-Derlin-1 complexes from HEK293A cells at 4 °C for 16 h. For the cell-based immunoprecipitation, compounds were added to the culture medium of HEK293A cells for 24 h.

**Chemical compounds.** Compound #56 (InterBioScreen Ltd), #56-20, and #56-26 (Vitas-M laboratory) were purchased. Other compounds were synthesized as described in Supplementary Methods.

**Permeability assay and metabolic stability assay.** The permeability and metabolic stability of the compounds were determined commercially using the CEREP in vitro drug absorption and in vitro drug metabolism service, respectively.

**Surface plasmon resonance.** SPR was analyzed with Biacore T100 (GE Healthcare). The biotinylated SOD1 (5-20) peptide and biotinylated Derlin-1 (CT4) peptide were captured as a ligand with Biotin CAPture Kit, Series S (GE Healthcare). #56-20 or #56-26 in a buffer containing 10 mM HEPES, pH 7.5, 150 mM NaCl, 3 mM EDTA, 0.005% Tween-20, and 0.5% DMSO were flowed as an analyte.

**FP analysis.** For FP analysis, 250 nM #56-20 was incubated with various concentrations of biotinylated SOD1 (5-20) peptide or biotinylated Derlin-1 (CT4) peptide in a buffer containing 10 mM HEPES, pH 7.5, 150 mM NaCl, 3 mM EDTA, and 0.005% Tween-20 for 6 h at room temperature in the 384-well plates (Greiner). The FP (405 nm excitation filter/535 nm emission filter) was measured using an ARVO X5 (Perkin Elmer). The experiment was repeated three times on three separate days. $K_d$ was calculated by the following equation[39]: $P = P_L + [(P_{RL} - P_L)/L_T] \times ½ [(L_T + R_T + K_d) - √ ((L_T + R_T + K_d)^2 - (4 \times R_T \times L_T))]$, where $P$ is the observed polarization value; $P_{RL}$ and $P_L$ are the polarization signals for fully bound and displaced ligand; and $L_T$ and $R_T$ are total concentrations of #56-20 and recombinant SOD1.

**Pull-down assay for target identification.** HEK293A cells transfected with vector, Flag-SOD1$^{G93A}$, Flag-SOD1$^{WT}$, or Derlin-1-Flag were treated with 200 µM PB56 for 24 h and then irradiated with 365-nm ultraviolet (UV) light for 1 h on ice. In the competition assay, HEK293A cells transfected with Flag-SOD1$^{G93A}$ were treated with 200 µM PB56 for 24 h with or without pre-treatment with 20 µM #56-59 for 12 h and were irradiated with 365-nm UV light for 1 h. Cells were lysed in a buffer containing 20 mM Tris-HCl, pH 7.5, 150 mM NaCl, 10 mM EDTA, 1% Triton X-100, 5 µg ml$^{-1}$ leupeptin, and 1 mM phenylmethylsulfonyl fluoride.

**Recombinant proteins and SOD activity assay.** The cDNAs encoding GST-SOD1$^{WT}$ and GST-SOD1$^{G93A}$ were constructed in the pGEX-6P-1 vector (GE Healthcare), and each plasmid was transformed into *Escherichia coli* (BL21). The transformed cells were cultured in Luria Broth medium containing 50 µg ml$^{-1}$ ampicillin at 37 °C for 16 h. Isopropyl β-ᴅ-1-thiogalactopyranoside was then added at a final concentration of 0.1 mM, and the cells were incubated at 30 °C for an additional 2 h. Cells were lysed with chilled lysis buffer (phosphate-buffered saline (PBS) containing 10 mM EDTA and 1% Triton X-100). After centrifugation at 17,700 × g, the supernatant was incubated with Glutathione Sepharose 4B (GE Healthcare) at 4 °C for more than 2 h. After three washes with PBS containing 10 mM EDTA, proteins were eluted using the elution buffer (50 mM Tris-HCl, pH 8.0, and 10 mM ʟ-glutathione). The proteins were incubated with 100 µM CuSO$_4$ and ZnCl$_2$, dialyzed using PBS, and incubated with DMSO and 50 µM #56-20 or #56-26 at 4 °C for 16 h. SOD activity was measured using a SOD Assay Kit

according to the manufacturer's instructions (Dojindo). The experiment was repeated three times on three separate days.

**Motoneuron survival assay using iPSCs.** Use of human iPSCs was approved by the Ethics Committees of the respective departments at Kyoto University. All methods were performed in accordance with approved guidelines. Informed consent was obtained from all subjects. Motoneurons were generated from previously established iPSCs by introducing transcription factors Lhx3, Ngn2, and Isl1 under the control of tetracycline[31]. Some iPSC clones, Control 1, ALS1, ALS2, and ALS4, are deposited into RIKEN BioResource Center (depository numbers are HPS0063, HPS0251, HPS0252, and HPS0292, respectively). The compound screen was previously established[31]. The iPSCs were dissociated to single cells using Accutase (Innovative Cell Technologies, San Diego, CA, USA) and plated onto Matrigel-coated 96-well plates (Corning, Corning, NY, USA) with neuronal medium (DMEM/F12 (Thermo Fisher Scientific), 100 μg ml$^{-1}$ apotransferrin (Sigma), 5 μg ml$^{-1}$ insulin (Sigma), 30 nM selenite (Sigma), 20 nM progesterone (Sigma), 100 nM putrescine (Sigma)) containing 1 μM retinoic acid (Sigma), 1 μM smoothened agonist), 10 ng ml$^{-1}$ brain-derived neurotrophic factor (R&D Systems, Minneapolis, MN, USA), 10 ng ml$^{-1}$ glial cell-derived neurotrophic factor (R&D Systems), and 10 ng ml$^{-1}$ NT-3 (R&D Systems) with 1 μg ml$^{-1}$ doxycycline (TAKARA), and cultured for 7 days. Compounds were added to cells on day 7, and cultures were continued until day 14. Cells were fixed and stained on days 7 and 14. The number of surviving motoneurons stained with βIII-tubulin (Covance, Princeton, NJ, USA) was quantified by IN Cell Analyzer 6000 (GE Healthcare, Chicago, IL, USA) and IN Cell Developer toolbox software 1.9 (GE Healthcare).

**Animals.** All experiments were performed in accordance with protocols approved by the Animal Research Committee of the Graduate School of Pharmaceutical Sciences, The University of Tokyo (Tokyo, Japan). The mouse genotypes were verified by PCR at weaning and before the administration of compounds. At 22 weeks of age, approximately 6 weeks before the usual onset timing as defined in our previous study[13], male mice with the human SOD1$^{G93A}$ transgene (G1L/+ line, backcrossed to C57BL/6) were randomly separated into three groups and continuously infused with DMSO as control, with 1 mM #56-40, or with 3 mM #56-59, using osmotic pumps (flow rate 0.15 μl h$^{-1}$, Alzet Mode 2006) by i.c.v. cannulation with the Brain Infusion Kit 3 (Alzet). Pumps were placed on the back of the mouse and were replaced every 6 weeks until the mouse showed paralysis. Onset was determined at the time point when the average of week's rotarod performance (three times per week) began to decline at an accelerated speed to 40 rpm for 5 min. The genotype of the mouse that did not show any symptoms was again confirmed by PCR, and the treatment trial of this mouse was stopped at 58 weeks of age. The investigators were not blinded to allocation during experiments and outcome assessment.

**Measurement of #56-59 concentration.** Male mice (C57BL/6) were continuously infused with 3 mM #56-59 using osmotic pumps by i.c.v. cannulation. The brain and spinal cord were extirpated at 1, 3, and 5 weeks after the start of the infusion and homogenized with phosphate buffer. The tissue homogenates were deproteinized with acetonitrile containing the internal standard (methyltestosterone) and centrifuged at 10,000 × g for 5 min at 4 °C. The supernatants were subjected to liquid chromatography-tandem triple quadrupole mass spectrometry analysis (LCMS-8060, Shimadzu, Japan) to quantify the concentration of #56-59 in the brain and spinal cord. The samples that were below the detection limit (Supplementary Table 2) were eliminated in the Supplementary Figure 8b.

**Nissl staining and immunohistochemistry.** Mice were perfused with PBS and then with 4% paraformaldehyde (PFA) in PBS and the spinal cords (L2–L5) were fixed with 4% PFA in PBS for 1 day. The lumbar spinal cords were then incubated in 30% sucrose for 24 h, embedded in CryoMount I (Muto PureChemicals), and frozen sections (40 μm) were produced. Nissl stained were performed using 0.1% cresyl violet acetate (WALDECK). Motoneurons showing a clear nucleolus and distinctly labeled cytoplasm with neuronal morphology, with cell bodies >20 μm in diameter and located in the anterior horn, were included in cell counts. All motoneuron counts were performed in a blinded fashion. For the immunohistochemistry, frozen sections were blocked with 1% BSA at room temperature, and each section was incubated with the primary antibodies. After three washes with PBS, sections were incubated with the secondary antibody solution at room temperature. All images were obtained using a Leica TCS SP5 confocal laser scanning microscope.

**Statistical analysis.** No statistical methods were used to predetermine sample size. Motoneurons survival assay were analyzed using one-way analysis of variance followed by Tukey's post hoc test (with similar variance) or Kruskal–Wallis followed by Dunn's test (#56-59-treated ALS1 with non-similar variance) to determine statistical significances of the data ($n = 6$; $p < 0.05$, post hoc test, *$p < 0.05$). Onset and survival were statistically analyzed using the Kaplan–Meier method followed by the log-rank test and the Gehan–Breslow–Wilcoxon test (*$p < 0.05$). To ensure randomization of spinal cord section for motoneurons count, the five frozen sections of L2 to L5 were randomly chosen (WT mice (34 weeks old): $n = 4$;

DMSO-treated and #56-59-treated SOD1$^{G93A}$ transgenic mice (31 weeks old): $n = 8$ per group, *$p < 0.05$: the significance with similar variance was evaluated using unpaired, two-tailed Student's $t$ test).

**Data availability.** The data that support the findings of this study are available from the corresponding author on reasonable request.

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

## Acknowledgements

We thank H. Nishitoh, H. Kadowaki, and N. Yamaguchi (at The University of Tokyo) for discussion and assistance; Y. Ikegaya (at The University of Tokyo) for the instruction of the intracerebroventricular cannulation; One-stop Sharing Facility Center For Future Drug Discoveries (at The University of Tokyo) for sharing the instrument; and H. Nishina (at Tokyo Medical and Dental University) for the support. We are grateful to K. Matsuzaki (at Nagoya Institute of Technology), N. Ieda, M. Kawaguchi (at Nagoya City University), T. Fukuyama and Y. Beniyama (at Nagoya University) for the support of chemical synthesis. A portion of this study resulted from "Understanding of molecular and environmental bases for brain health" performed under the Strategic Research Program for Brain Sciences by the Ministry of Education, Culture, Sports, Science, and Technology of Japan (MEXT) (to H.I.). This study was also supported by Grants-in-Aid for Scientific Research from the Japanese Society for the Promotion of Science and MEXT (20229004 and 25221302 to H.I., 26650028 and 16K18513 to K.H.), the Advanced research for medical products Mining Program of the National Institute of Biomedical Innovation (to H.I.), Project for Elucidating and Controlling Mechanisms of Aging and Longevity from Japan Agency for Medical Research and Development (AMED) (JP17gm5010001 to H.I.), as well as the Platform for Drug Discovery, Informatics, and Structural Life Science from AMED (JP17am0101086 to H.K.). This research is partially supported by Platform Project for Supporting Drug Discovery and Life Science Research (Basis for Supporting Innovative Drug Discovery and Life Science Research (BINDS)) (JP17am0101087).

## Author contributions

N.T. and K.H. performed most of the experiments. K.I. and H.I. performed the iPSCs experiments. T.H., A.B., H.Y., N.S., S.N., H.N., S.-i.I., N.U., N.K., S.Y., and M.S. designed and synthesized the chemical compounds. H.I. conceived the idea. N.T., K.H., T.F., and H.I. designed the experiments with critical assistance from M.S., H.K., T.O., T.N., I.N., N. T., K.H., and H.I. mainly wrote the manuscript. All authors commented on the manuscript.

## Additional information

**Competing interests:** The University of Tokyo and Public University Corporation Nagoya City University are applying for a patent of #56 analogs. The inventors are N.T., K.H., M.S., H.K., T.O., T.N., T.F., H.I., N.S., T.H., S.N., H.N., and S.i.I. The application number is PCT/JP2017/021895 and the status is patent-pending. The patent covers the effect of #56-59 analogs to ALS pathology in model mouse. The other authors declare no competing interests.

