## [Peer Review File · Nature Communications]

Reviewers' comments:

Reviewer #1 (Remarks to the Author):

The manuscript "A small-molecule inhibitor of SOD1-Derlin-1 interaction ameliorates ALS pathology" by Tsuburaya et al describe a small molecule screen and identification of hits that rescue cellular and animal models of ALS.

The interaction of mutant SOD1 with Derlin-1 has previously been shown to be pathological and therefore such protein-protein disruptors could potentially be therapeutic. The authors have hypothesized that disruption of mutant SOD1-Derlin-1 would reduce ER stress and increase the survival of patient neurons as well as animal models of ALS expressing mutant SOD1. The authors screened 160,000 small molecules on a FRET-based assay to identify hits that inhibit SOD1-Derlin-1 interaction. They identified hits and obtained additional analogs that were tested in in vitro IP cell-based assays. A few of the best analogs were tested on patient iPSC-motor neurons and later in the SOD1 G93A ALS animal model.

In general, the data in the manuscript is well organized and written clearly. It would be good if the authors could address the following points:

It looks like the primary screen was done using the FRET assay but the analogs were only tested in the in vitro IP assay. What is the relationship between the FRET assay and the IP cell based assay? How do the analogs compare to the other hits when tested side by side in the FRET assay? How does inhibition in the FRET assay translate to inhibition in the cell-based assay? If potent inhibitors identified in the FRET assay do not show inhibition in the IP assay, could that solely be attributed to cell permeability? Are the drugs metabolized by the cell line or could they bind to proteins in the culture media?

Have the authors conducted any in vitro or in vivo drug metabolism studies?

How stable are the drugs in vitro in the cell culture (cell lines and iPSC-neurons) and in vivo?

What are the solubility properties of the small molecules?

In the experiments with iPSC-neurons, the authors measured day14/day7 % survival. However, this measurement can be confounded by motor neuron progenitors that exist in differentiating motor neuron cultures and continue to produce neurons throughout the culture. How can the authors be sure that the effect is due to increased survival and not increased generation of new neurons from progenitors that are in the cultures? Based on these experiments, one cannot conclude that the compounds improve the survival of the iPSC-neurons.

Do the patient neurons show disease-relevant survival defect phenotype when compared with neurons from healthy individuals? Is this phenotype due to SOD1 mutation? Have the authors produced isogenic control line to correct the SOD1 mutation to show that the phenotype is directly linked to the SOD1 mutation? Is the phenotype due to interaction of the endogenous mutant SOD1 with Derlin-1? These data need to be provided independent of the small molecule studies.

How were the iPSC quality controlled? Did they have normal karyotypes? What percent motor neurons were present in the cultures? Is there any pluripotency data around the iPSC lines that the data can provide to confirm that the iPSC lines were of high quality? Minimum standard Q/C measures would include karyotyping, pluripotency markers, differentiation into germ layers, confirmation of the mutation by sequencing, and motor neuron markers of the iPSC-neurons.

In the in vivo efficacy studies the authors used an osmotic mini pump to deliver the drug. However, no prior studies were carried out to make sure the molecules make it out of the pump and are stable in the animals and reach the target cells. Are the drugs soluble in the animals? How quickly do they get metabolized? What are the PK/PD properties? Do the metabolites have any activity? Do the drugs inhibit mutant SOD1-Derlin1 interaction in the motor neurons of the SOD1 animals?

What effect do the compounds have on ER health of iPSC-neurons and the neurons in the mutant SOD1 animals that were treated?

Reviewer #2 (Remarks to the Author):

In the present manuscript Tsuburaya et al. focus on previously published interaction between mutant SOD1 and Derlin-1, important component of ER-associated degradation (ERAD) machinery (Nishitoh et al., Genes Dev 2008). As their previous findings suggested that mutSOD1-Derlin-1 interaction is important for mutSOD1-mediated toxicity authors set out to discover a small molecule inhibitor of this interaction. Moreover, they test discovered inhibitors for neuroprotective effects in human iPSC-derived motor neurons and in SOD1-G93A mouse ALS model.

Overall this manuscript is novel and of interest to the field. Authors discovered interesting and potent small molecules for potential use in treatment of ALS. In my eyes, this paper consists of 2 parts:

- the first part focuses on small molecule discovery and characterization. This part would need minor experiments and clarifications (suggested below) to further strengthen authors's claims.
- the second part focuses on showing the effects of the discovered small molecules on human iPSC-derived motor neurons and SOD1-G93A mouse model. In my opinion, this part need major improvements (most importantly using appropriate controls and increasing the number of cell lines). Given group's previous work that this manuscript builds upon, authors should test particular mechanisms previously suggested (namely Ask1-dependent cell death pathway) to further strengthen their claim for potential use of compound #56 analogs in ALS.

Specific comments:

Authors successfully established a high-throughput screening (HTS) based on FRET-based detection of SOD1-Derlin-1 interactions. As the first step a total of 160,000 small molecules was screened for inhibition of FRET signal and 1,460 compounds decreased FRET signal for

at least 15%. In the second step, authors repeated the screen with only the 1,460 compounds, but also eliminating false positives by using monomolecular FRET assay as control. After this filtering, 131 compounds showed at least 15% inhibition of the SOD1-Derlin-1-derived FRET signal.

All of the 131 compounds were further tested in an in vitro co-immunoprecipitation assay and 12 compounds had a negative effect of co-immunoprecipitation of SOD1 and Derlin-1. Out of these 12 compounds, authors chose to focus on compound #56, because of its dose-dependency in FRET and "drug-like chemical structure".

- What does "drug-like chemical structure" exactly mean? Could authors explain this?
- How about compound #12? Was #12 not chosen because it is not dose dependent and doesn't have a "drug-like chemical" structure?

In the next experiments authors chose to test commercially available analogs of compound #56. Even though two of them showed lower IC50 values than the original #56, unfortunately none of them was cell permeable.

- Does the original #56 cell penetrate into cells? Authors should test and report this.

Given the non-cell-permeability of commercial analogs, authors synthesized additional analogs and found several of them that were potent and cell-permeable, notably #56-59 and #56-111.

Authors then went back to non cell-permeable #56-20 and #56-26 to test the mechanism of SOD1-Derlin-1 inhibition.

- Why did authors perform these experiments with #56-20 and #56-26? It would be more relevant to perform them with cell-permeable compounds used in experiments in Figure 6 (particularly #56-59).

In the final Figure 6 authors test the effects of #56 compounds on iPSC-derived human motor neurons and SOD1-G93A mouse model.

#56-40 and #56-59 were applied to only 1 patient iPSC-derived MN line. This should be expanded to at least 3 patient lines. In addition, these experiments lack WT control and it is not clear whether the increase in MN survival is specific to mutSOD1 effects or the compounds simply ameliorate survival of MNs in in vitro culture conditions.

- Therefore, at minimum authors should expand these experiments with additional 2 mutant SOD1 lines and minimum of 2 wild type lines.
- Of potential interest would be to also use, mutant TDP43-carrying lines (particularly because this possibility is mentioned in the discussion) and lines from a different neurodegenerative disease (for example Parkinson's disease) to further test the ALS-specific effect of #56 compounds.

In further experiments authors tested the effects of #56 compounds in SOD1-G93A mouse model. These results show an impressive effect on both disease onset and survival with #56-59.

- Is n=12 in these experiments? Could authors state that clearly in the figure or figure legend?
- Also, in the methods section authors describe that these experiments were not done in a blinded manner. Particularly when measuring disease onset, I believe it would be important to repeat these experiments in a blinded manner (or at least re-analyze them in blinded

manner if that is possible).

- In their Nishitoh et al., Genes Dev 2008 publication, they showed that mutSOD1-Derlin-1 results in activation of ASK1-dependent cell death. Authors should explore this mechanism in tissues of their treated mice: is ASK1 activation altered with compound #56-59 administration?

- Authors should also test their compound on primary motor neurons from Ask1 -/- mice in order to show compound specificity and confirm the mechanism they suggested.

Ana Jovicic, PhD.
Genentech, Inc.

Reviewer #3 (Remarks to the Author):

This is an interesting paper that describes the identification of SOD1/Derlin1 PPI inhibitors. The authors detail an HTRF assay used to screen a library of small molecules. The identified molecules are assessed mainly in cell-based assays. The leading scaffolds are tested in vivo and show some activity.

It is clear that a lot of work has been conducted to get to this stage and the authors should be commended for that. However, the paper suffers from several shortcomings that should be addressed prior to publication here or elsewhere.

1. Lack of in vitro characterisation is a major shortcoming of the work. It is surprising to see no in vitro quantification of the affinities of the reported molecules. This is critical information for any new inhibitor being reported. The lack of this data is particularly surprising given that the authors have access to recombinant proteins. The authors should conduct one or more of: SPR, ITC, FP, or MST to determine the binding affinity of their leading molecules (#56, 56-40, #56-59).

2. Why have biotinylated SOD1 and Derlin-1 been used in the FP assays? The authors should have used the GST-tagged proteins they produced for the SOD activity assay. That way, they could have obtained a binding constant for the compounds. From the graph presented (Fig. 5b), the k_d of their leading molecule looks very high ($\sim 50\mu\text{M}$).

3. The pull-down experiments with the photo-biotin-PEG3-#56-59 are flawed, in that the plasmids have been introduced to overexpress the target proteins are being overexpressed in the cells.

4. Figure 4 should be moved to SI. It is just a series of western blots and the conclusion drawn from it is straightforward. If anyone is interested in seeing the data, they can look in the SI.

6. the authors present the hypothesis that #56-59 selectively binds mutant SOD and so is superior (potentially) to siRNA-based therapies currently in trials. I would like to see some

in vitro verification of the asserted selectivity, using one of the methods listed in point 1 above. They don't need to test all the mutants in figure 4, but maybe 10 or so showing the most extreme ranges of activity (highest and lowest). This would not only back up their IP data with K_d values for the target, it would also provide evidence for their hypothesis.

[Response to the reviewer]

We appreciate reviewers' interest in our work, the detailed evaluations and the helpful suggestions. Those comments are all valuable and very helpful for revising and improving our paper. We hereby address the reviewers' concerns as below and made substantial changes in the manuscript.

Reviewer #1:

The manuscript “A small-molecule inhibitor of SOD1-Derlin-1 interaction ameliorates ALS pathology” by Tsuburaya *et al* describe a small molecule screen and identification of hits that rescue cellular and animal models of ALS.

The interaction of mutant SOD1 with Derlin-1 has previously been shown to be pathological and therefore such protein-protein disruptors could potentially be therapeutic. The authors have hypothesized that disruption of mutant SOD1-Derlin-1 would reduce ER stress and increase the survival of patient neurons as well as animal models of ALS expressing mutant SOD1. The authors screened 160,000 small molecules on a FRET-based assay to identify hits that inhibit SOD1-Derlin-1 interaction. They identified hits and obtained additional analogs that were tested in *in vitro* IP cell-based assays. A few of the best analogs were tested on patient iPSC-motor neurons and later in the SOD1 G93A ALS animal model.

In general, the data in the manuscript is well organized and written clearly. It would be good if the authors could address the following points:

It looks like the primary screen was done using the FRET assay but the analogs were only tested in the *in vitro* IP assay. What is the relationship between the FRET assay and the IP cell based assay? How do the analogs compare to the other hits when tested side by side in the FRET assay? How does inhibition in the FRET assay translate to inhibition in the cell-based assay? If potent inhibitors identified in the FRET assay do not show inhibition in the IP assay, could that solely be attributed to cell permeability? Are the drugs metabolized by the cell line or could they bind to proteins in the culture media?

Response: This is an important point. There are correlations between the FRET assay and the *in vitro* IP assay (**Fig. 3a, b, Supplementary Fig. 3c**), but not between the FRET and the cell-based IP assay nor between the *in vitro* IP assay and the cell-based IP assay. The compounds that show high inhibition activity in FRET assay do not necessarily inhibit the interaction in cells, and some compounds that possess potent inhibition activity in cells have little effect in FRET assay. To evaluate the possibility that the undesired interaction in culture medium is the cause of the inactivity, we examined the effect of #56 analogs (including potent *in vitro* inhibitors, #56-20 and #56-26) in the serum-depleted medium.

As shown in newly added **Supplementary Figure 3a**, some of the compounds showed the inhibition activity in the absence of serum, suggesting that the interaction with serum-derived substances in culture medium is one of the reasons for the inability of inhibiting the interaction in cell-based assay. In addition, the permeability data performed in Caco-2 cells suggested that the compound that could not show the inhibitory activities in cells (#56-26) have low permeability compared to the active compound (#56-40), indicating that the low permeability of the plasma membrane is also one of the causes (**Supplementary Figure 3b**). Since the stability assay with liver microsomes showed the relatively stable character of #56-26 rather than #56-40 (**Supplementary Figure 3b**), we speculate that their undesired interaction in culture medium and low permeability may be the critical reason for the discrepancy between the FRET (*in vitro* IP) assay and the cell-based IP assay. But the possibility that modifications of compounds also suppress the activity could not be eliminated as the reviewer pointed out. We improved the corresponding part of the manuscript (p. 12, lines 3-9).

Have the authors conducted any in vitro or in vivo drug metabolism studies?

How stable are the drugs in vitro in the cell culture (cell lines and iPSC-neurons) and in vivo?

What are the solubility properties of the small molecules?

Response: As mentioned above, here we conducted the stability assay with liver microsome (**Supplementary Figure 3b**). #56-40 showed a relatively unstable property. Thus, although the cell lines, iPSC-neurons or the cells in CNS do not express the typical metabolizing enzymes compared to hepatic cells, #56-40 (and probably also #56-59) could be metabolized. As discussed below, it is possible that the modified #56-40 and #56-59 were active in the cells and *in vivo*.

In the experiments with iPSC-neurons, the authors measured day14/day7 % survival. However, this measurement can be confounded by motor neuron progenitors that exist in differentiating motor neuron cultures and continue to produce neurons throughout the culture. How can the authors be sure that the effect is due to increased survival and not

increased generation of new neurons from progenitors that are in the cultures? Based on these experiments, one cannot conclude that the compounds improve the survival of the iPSC-neurons.

Response: We fully agree with the reviewer. Related to reviewer #2's concern, we assayed the effect of #56-59 to healthy control-derived iPS motoneurons (**Supplementary Fig. 8a**). The addition of #56-59 did not improve the survival ratio of the control iPS-motoneurons, suggesting that they do not facilitate the differentiation of progenitor cells or the general survival of motoneurons independent of ALS pathology.

Do the patient neurons show disease-relevant survival defect phenotype when compared with neurons from healthy individuals? Is this phenotype due to SOD1 mutation? Have the authors produced isogenic control line to correct the SOD1 mutation to show that the phenotype is directly linked to the SOD1 mutation? Is the phenotype due to interaction of the endogenous mutant SOD1 with Derlin-1? These data need to be provided independent of the small molecule studies.

How were the iPSC quality controlled? Did they have normal karyotypes? What percent motor neurons were present in the cultures? Is there any pluripotency data around the iPSC lines that the data can provide to confirm that the iPSC lines were of high quality? Minimum standard Q/C measures would include karyotyping, pluripotency markers, differentiation into germ layers, confirmation of the mutation by sequencing, and motor neuron markers of the iPSC-neurons.

Response: This is also an important point. The iPSCs and the assay system were generated in our previous work (reference 31). Minimum standard Q/C measures of the iPSC lines utilized in the present study and the characterization of SOD1^{mut}-dependent motoneuron death, including the isogenic control of ALS 1, were also investigated in that study. Since our original manuscript was ambiguous, we improved the corresponding parts (p. 17, lines 4-5 and p. 34, lines 12-13).

In the in vivo efficacy studies the authors used an osmotic mini pump to deliver the drug. However, no prior studies were carried out to make sure the molecules make it out of the

pump and are stable in the animals and reach the target cells. Are the drugs soluble in the animals? How quickly do they get metabolized? What are the PK/PD properties? Do the metabolites have any activity? Do the drugs inhibit mutant SOD1-Derlin1 interaction in the motor neurons of the SOD1 animals?

Response: We fully agree with the reviewer. To evaluate the property of the #56-59 *in vivo*, we calculated the concentration of the #56-59 in the brain and spinal cord at 1, 3, 5 weeks after the start of continuous infusion. As shown in **Supplementary Figure 8b** and **Supplementary Table 2**, #56-59 reached the brain and spinal cord. However, the concentration of the compound was far lower than the concentration used in the experiment with cell culture, even if we took account of the unbound fractions of #56-59 within serum (**Supplementary Fig. 8c**). It could be due to the modification of #56-59 as the reviewer suggested. Since #56-59 could be modified as discussed above, it is possible that modified #56-59 that we cannot identify in the present study might contribute to the inhibition of SOD1-Derlin-1 interaction in our experiments. The difference of the treatment time course could also be a possibility. In the long-term treatment, #56-59 could occupy the DBR of newly synthesized SOD1^{mut} before its interaction with Derlin-1. Thus, #56-59 would be able to efficiently inhibit the SOD1-Derlin-1 interaction compared to the cell-based immunoprecipitation assay, where #56-59 might dissociate from the already formed complexes. It is also possible that the continuous infusion over 5 weeks would reach higher concentration, because the concentration of #56-59 was increased in spinal cord at 5 weeks of infusion (**Supplementary Fig. 8b, Supplementary Table 2**). We discussed about this point in the revised manuscript (pp. 21, line 1 - pp. 22, line 4). Although we also tried to evaluate the inhibition activity *in vivo*, the level of the SOD1-Derlin-1 interaction varies even in the non-treated ALS model mice and it seems to be technically difficult to significantly evaluate the efficacy of #56-59 in the animals (**data below**).

The lysates of spinal cord were analyzed by IP-IB with the indicated antibodies. SOD1 G93A Tg mice (33-34 weeks old).

What effect do the compounds have on ER health of iPSC-neurons and the neurons in the mutant SOD1 animals that were treated?

Response: We are also interested in this point. We assayed the expression level of ER stress markers with qRT-PCR in the lumbar spinal cord at 31 weeks old, when the onset of DMSO-treated mouse and difference in the number of motoneurons was detected. However, we could not observe the clear induction of these markers even in DMSO-treated ALS model mice (**data below**). We assumed that the presence of glial cells masked the upregulation of these genes in motoneurons. Next, we investigated the phosphorylation of eIF2 α in the spinal cord sections. This marker was chosen because the previous study showed that eIF2 α was phosphorylated in motoneurons at the onset of ALS model mice (Sun, S. et al. *Proc. Natl. Acad. Sci. USA* 2015). However, we could not detect the apparent difference even between the WT and SOD1 G93A Tg mice (**data below**), probably because of the difference of the mouse lines. Also, there was no difference between DMSO- and #56-59-treated mice (**data below**). Thus, we failed to evaluate the effect of our compounds on ER health in this condition. Related to the reviewer #2's question, it is our important future subject to reveal when and what signals are inhibited or activated under the

inhibition of SOD1-Derlin-1 interaction in our treatment protocol. We discussed about this point in the improved manuscript. (p. 23, lines 3-13).

ER stress induction in ALS model mice.

(a) Gene expression analysis for mice spinal cord of WT, DMSO-treated SOD1 G93A Tg and #56-59-treated SOD1 G93A Tg using qRT-PCR. WT mice: 34 weeks old n=4, DMSO-treated SOD1 G93A Tg mice: 31 weeks old n=5, #56-59-treated SOD1 G93A Tg mice: 31 weeks old n=6.

(b) The phosphorylation of eIF2 α in motoneurons of WT (34 weeks old) and SOD1 G93A Tg mice (33-34 weeks old). Three sections per mouse. WT mice: n=3, SOD1 G93A Tg: n=3. Scale bar = 25 μ m.

(c) The phosphorylation of eIF2 α in motoneurons of DMSO-treated and #56-59-treated SOD1 G93A Tg mice. Four sections per mouse. DMSO-treated SOD1 G93A Tg: n=7, #56-59-treated SOD1 G93A Tg: n=8. Scale bar = 50 μ m.

Reviewer #2:

In the present manuscript Tsuburaya et al. focus on previously published interaction between mutant SOD1 and Derlin-1, important component of ER-associated degradation (ERAD) machinery (Nishitoh et al., Genes Dev 2008). As their previous findings suggested that mutSOD1-Derlin-1 interaction is important for mutSOD1-mediated toxicity authors set out to discover a small molecule inhibitor of this interaction. Moreover, they test discovered inhibitors for neuroprotective effects in human iPSC-derived motor neurons and in SOD1-G93A mouse ALS model.

Overall this manuscript is novel and of interest to the field. Authors discovered interesting and potent small molecules for potential use in treatment of ALS. In my eyes, this paper consists of 2 parts:

- the first part focuses on small molecule discovery and characterization. This part would need minor experiments and clarifications (suggested below) to further strengthen authors' claims.

- the second part focuses on showing the effects of the discovered small molecules on human iPSC-derived motor neurons and SOD1-G93A mouse model. In my opinion, this part need major improvements (most importantly using appropriate controls and increasing the number of cell lines). Given group's previous work that this manuscript builds upon, authors should test particular mechanisms previously suggested (namely Ask1-dependent cell death pathway) to further strengthen their claim for potential use of compound #56 analogs in ALS.

Specific comments:

Authors successfully established a high-throughput screening (HTS) based on FRET-based detection of SOD1-Derlin-1 interactions. As the first step a total of 160,000 small molecules was screened for inhibition of FRET signal and 1,460 compounds decreased FRET signal for at least 15%. In the second step, authors repeated the screen with only the 1,460 compounds, but also eliminating false positives by using monomolecular FRET assay as control. After this filtering, 131 compounds showed at least 15% inhibition of the SOD1-Derlin-1-derived FRET signal.

All of the 131 compounds were further tested in an in vitro co-immunoprecipitation assay

and 12 compounds had a negative effect of co-immunoprecipitation of SOD1 and Derlin-1. Out of these 12 compounds, authors chose to focus on compound #56, because of its dose-dependency in FRET and “drug-like chemical structure”.

- What does “drug-like chemical structure” exactly mean? Could authors explain this?
- How about compound #12? Was #12 not chosen because it is not dose dependent and doesn't have a “drug-like chemical” structure?

Response: As the reviewer pointed out, “drug-like chemical structure” was not defined. We selected these compounds according to the Lipinski's rule of five, the absence of reactive functional groups, and the exclusion of promiscuous hitters. The easiness to speculate the pharmacophore and to synthesize the analogs was also taken into consideration. To define “drug-like chemical structure”, we improved the part of the manuscript (pp. 11, lines 15 – pp.12, line 6).

The compound #12 was eliminated because of the low dose-dependency (**data below**).

The results of compound #12 in the 3rd screening. Blue: inhibition (%) against FRET signal generated by Flag-SOD1^{G93A} and Derlin-1-HA, red: inhibition (%) against FRET signal generated by HA-SOD1^{WT}-Flag.

In the next experiments authors chose to test commercially available analogs of compound #56. Even though two of them showed lower IC50 values than the original #56, unfortunately none of them was cell permeable.

- Does the original #56 cell penetrate into cells? Authors should test and report this.

Response: This is an important point. Related to the Reviewer #1's first question, the

permeability data showed that the compounds that could not show the inhibitory activities in cells (#56-26) have low permeability compared to the active compound (#56-40) (**Supplementary Figure 3b**). Although we did not test the original #56, we speculate that the inabilities of #56 and its analogs (including #56-26) to inhibit SOD1-Derlin-1 interaction are largely due to the low permeability of the plasma membrane or undesirable interaction in culture medium.

Given the non-cell-permeability of commercial analogs, authors synthesized additional analogs and found several of them that were potent and cell-permeable, notably #56-59 and #56-111.

Authors then went back to non cell-permeable #56-20 and #56-26 to test the mechanism of SOD1-Derlin-1 inhibition.

- Why did authors perform these experiments with #56-20 and #56-26? It would be more relevant to perform them with cell-permeable compounds used in experiments in Figure 6 (particularly #56-59).

Response: This is also an important point. To reveal the mechanism of #56-mediated PPI inhibition and identify the direct target of #56 analogs, it is important to evaluate the interaction *in vitro*. However, the compounds that are active in cell-based IP assay are not necessarily potent inhibitors in *in vitro* assay and we could detect only a weak interaction between SOD1 (5-20) peptide and #56-59 in fluorescence polarization (FP) assay even in the presence of 100 μ M peptide (Related to the reviewer #1's question). Thus, we utilized the #56-20 and #56-26, which efficiently inhibited the interaction in the *in vitro* IP assay, instead of #56-59 to clearly demonstrate the direct interaction between the #56 analogs and target protein in *in vitro* assays. In addition, we synthesized PB56 to confirm the interaction with SOD1 within the cells (**Figure 4c-e**). Our manuscript was elusive in this point, and we improved the manuscript (pp. 14, line 15 – pp. 15, line 1 and p. 15, line 13-15).

In the final Figure 6 authors test the effects of #56 compounds on iPSC-derived human motor neurons and SOD1-G93A mouse model.

#56-40 and #56-59 were applied to only 1 patient iPSC-derived MN line. This should be

expanded to at least 3 patient lines. In addition, these experiments lack WT control and it is not clear whether the increase in MN survival is specific to mutSOD1 effects or the compounds simply ameliorate survival of MNs in in vitro culture conditions.

- Therefore, at minimum authors should expand these experiments with additional 2 mutant SOD1 lines and minimum of 2 wild type lines.

- Of potential interest would be to also use, mutant TDP43-carrying lines (particularly because this possibility is mentioned in the discussion) and lines from a different neurodegenerative disease (for example Parkinson's disease) to further test the ALS-specific effect of #56 compounds.

Response: We fully agree with the reviewer. Related to Reviewer #1's concern, we additionally assayed the effect of #56-59 to healthy control- and ALS patient-derived iPS motoneurons. As shown in **Supplementary Fig. 8a**, there were no increase in survival of control motoneurons, suggesting that #56-59 did not affect the general viability of motoneurons. It is important that we could observe the improved viability of another ALS patient-derived iPS motoneuron with *SOD1* mutation (ALS 2), while ALS 3 did not show the difference. Moreover, the toxicity of TDP-43 mutant was not ameliorated with #56-59 (ALS 4). Because the toxicity of #56-59 was enhanced in ALS 4 compared to the control motoneurons, we estimated that ALS motoneurons were feeble and the effective dose became narrow. This vulnerability might differ from each iPSC lines and explain the different response of iPS motoneurons with *SOD1* mutation. Since we could not deny the possibility that this vulnerability masked the positive effect of #56-59 in ALS 4, it is important to evaluate the involvement of the SOD1-Derlin-1 interaction by using multiple inhibitors, different iPSC-derived motoneurons and/or other methods..

In further experiments authors tested the effects of #56 compounds in SOD1-G93A mouse model. These results show an impressive effect on both disease onset and survival with #56-59.

- Is n=12 in these experiments? Could authors state that clearly in the figure or figure legend?

Response: We improved the part of the manuscript (p. 51, lines 13-14).

- Also, in the methods section authors describe that these experiments were not done in a blinded manner. Particularly when measuring disease onset, I believe it would be important to repeat these experiments in a blinded manner (or at least re-analyze them in blinded manner if that is possible).

Response: We determined the onset at the time point when the average of week's rota-rod performance (3 times per week) began to decline. Although we understand the reviewer's concern, we believe it would be acceptable. We improved the corresponding part of the manuscript (p. 36, lines 10-12).

- In their Nishitoh et al., Genes Dev 2008 publication, they showed that *mutSOD1-Derlin-1* results in activation of ASK1-dependent cell death. Authors should explore this mechanism in tissues of their treated mice: is ASK1 activation altered with compound #56-59 administration?

- Authors should also test their compound on primary motor neurons from *Ask1* *-/-* mice in order to show compound specificity and confirm the mechanism they suggested.

Response: We are also interested in this point. The depletion of ASK1 did not show the delay of onset, instead it prolonged the duration. This was distinct from the phenotype observed with the inhibition of SOD1-Derlin-1 interaction in this paper. Moreover, the administration of ASK1 inhibitors showed the reduction of the glial cells activity (reference 33), which is known to contribute to the duration of the pathology (reference 34, 35), while the #56-59 treatment did not show significant suppression of gliosis (**Supplementary Fig. 9**). Therefore, we assume that ASK1 is one of the important targets of SOD1-Derlin-1 interaction-mediated ER stress especially involved in the activation of glial cells, however, other signaling pathways that primarily contribute to the onset were also evoked by the SOD1-Derlin-1 interaction and these pathways were mainly inhibited in the present condition. It is our next interest to reveal the detailed signaling pathways affected in the present treatment protocol and the role of ASK1 in non-cell autonomous motoneuron death, as well as the effects of #56-59 on ASK1 pathway. We discussed about this point in the improved manuscript (p. 23, lines 3-13).

Reviewer #3:

This is an interesting paper that describes the identification of SOD1/Derlin1 PPI inhibitors. The authors detail and HTRF assay used to screen a library of small molecules. The identified molecules are assessed mainly in cell-based assays. The leading scaffolds are tested in vivo and show some activity.

It is clear that a lot of work has been conducted to get to this stage and the authors should be commended for that. However, the paper suffers from several shortcomings that should be addressed prior to publication here or elsewhere.

1. Lack of in vitro characterisation is a major shortcoming of the work. It is surprising to see no in vitro quantification of the affinities of the reported molecules. This is critical information for any new inhibitor being reported. The lack of this data is particularly surprising given that the authors have access to recombinant proteins. The authors should conduct one or more of: SPR, ITC, FP, or MST to determine the binding affinity of their leading molecules (#56, 56-40, #56-59).

Response: We fully agree with the reviewer. We calculated the binding affinity of #56-20 to recombinant full-length SOD1 in the FP assay because of its potent activity in the *in vitro* assays (**Supplementary Fig. 7a**). We also recognize the importance of measuring the K_d value of #56-59. However, related to the reviewer #1's question, we could detect only a weak interaction between SOD1 (5-20) peptide and #56-59 in fluorescence polarization (FP) assay even in the presence of 100 μM peptide and failed to calculate the K_d value of #56-59 because of the limitation of solubilized peptide concentration.

In addition, we also tried the SPR method with Biacore T100 (GE Healthcare) to calculate the K_d value of #56 analogs. We used the biotinylated peptide as a ligand and #56 analogs as an analyte. However, the interaction could not be detected probably due to the weak interaction and low solubility of the compounds. Although we agree with the importance of the K_d value of the compounds that are active in cell-based assays, for now it is technically very difficult and further optimizations of compounds are required as discussed below.

2. Why have biotinylated SOD1 and Derlin-1 been used in the FP assays? The authors should have used the GST-tagged proteins they produced for the SOD activity assay. That way, they could have obtained a binding constant for the compounds. From the graph presented (Fig. 5b), the k_d of their leading molecule looks very high (~50 μ M?).

Response: This is an important point. We used SOD1 and Derlin-1 peptide to identify the direct interaction site and predict the mechanism of PPI inhibition. However, the binding affinity should be evaluated with full-length SOD1 as the reviewer suggested. Thus, as mentioned above, we calculated the binding affinity of #56-20 to full-length recombinant SOD1 (**Supplementary Fig. 7a**). As the reviewer concerned, the K_d value was very high (81.1 μ M). It is possible that exposed DBR may be disordered, or the Derlin-1-binding structure may be less stable than the nonbinding structure in recombinant SOD1 as discussed in our previous study (reference 14) and, therefore, the K_d value became very high. However, the interaction with SOD1-peptide was also very weak. Thus, we agree that the affinities of these compounds were weak and further improvements are required. We added the discussion of this point in the revised manuscript (p. 24, lines 2- 6).

3. The pull-down experiments with the photo-biotin-PEG3-#56-59 are flawed, in that the plasmids have been introduced to overexpress the target proteins are being overexpressed in the cells.

Response: Although we completely agree with the reviewer, this experiment should be technically difficult because of the relatively weak inhibitory activity of PB56 and limited source of endogenous SOD1^{mut} expression. Because PB56 preferentially interacted with SOD1^{G93A} rather than SOD1^{WT}, we need to conduct this experiment in patients iPSCs. But as shown in **Figure 5a** and **Supplementary Fig. 8a**, iPSC-derived motoneurons were very sensitive to the compounds compared to the HEK293A cells. Taken the high K_d value of #56 analogs into account, the specificity of them in cells or *in vivo* should be carefully interpreted. The discussion about this point was added in the revised manuscripts (p. 24, lines 2- 6).

4. *Figure 4 should be moved to SI. It is just a series of western blots and the conclusion drawn from it is straightforward. If anyone is interested in seeing the data, they can look in the SI.*

Response: We agree with the reviewer. We transferred the previous figure 4 to **Supplementary Figure 5**.

6. *the authors present the hypothesis that #56-59 selectively binds mutant SOD and so is superior (potentially) to siRNA-based therapies currently in trials. I would like to see some in vitro verification of the asserted selectivity, using one of the methods listed in point 1 above. They don't need to test all the mutants in figure 4, but maybe 10 or so showing the most extreme ranges of activity (highest and lowest). This would not only back up their IP data with Kd values for the target, it would also provide evidence for their hypothesis.*

Response: We understand the reviewer's concern. As mentioned above, #56 analogs have the selectivity to SOD1^{mut}, but the specificity of them are unclear because of the high Kd value. Through this revision, we now think that it is unsuitable to compare with the siRNA-based therapies in the present manuscript. The corresponding description was deleted in the current one.

Additional Reference

Sun, S. et al. Translational profiling identifies a cascade of damage initiated in motor neurons and spreading to glia in mutant SOD1-mediated ALS. *Proc. Natl. Acad. Sci. USA* **112**, E6993–E7002 (2015).

REVIEWERS' COMMENTS:

Reviewer #1 (Remarks to the Author):

The authors have carried out additional experiments and provide new data to address the comments of the three reviewers. For example, the stability of the compounds has been evaluated. They have also tested additional patient iPSC lines. They have added significant explanation in the body of the manuscript as well. There were some data included in the rebuttal letter that is not in the manuscript. In my opinion, it may help to add the data into the supplementary section.

Reviewer #2 (Remarks to the Author):

As stated in the first round of the reviews, this paper presents novel and relevant findings. In addition, substantial amount of work has gone into this manuscript. In the resubmission, authors have improved the manuscript and removed some of the ambiguities that were present before. Importantly, they have answered our experimental questions. Among the additional experiments, they have added additional iPSC-derived MN lines and showed that #56 does not have an effect on the survival of control WT lines or mutant TDP43 line. Instead, compound shows a pro-survival effect in 2 out of 3 mutant SOD1 lines. Importantly, the compound shows increased survival in vivo in mutant SOD1 mouse model. I recommend this paper for publication in the present form.

Reviewer #3 (Remarks to the Author):

The authors have answered my questions (without providing any additional data), however, the disconnect between the in vitro and cell-based data, and the absence of any in vitro binding assay (or cellular thermal shift to show target engagement in cells) continues to be of concern. The authors should state in the manuscript text that binding of the molecules to SOD1 could not be detected by SPR. And the statements about

Given these shortcomings, I am unsure that the conclusions in the manuscript are supported by the data.

there are grammatical errors throughout the manuscript e.g. P.24 line 347, sentence begins with #56 analogue...

[Response to the reviewer]

REVIEWERS' COMMENTS:

Reviewer #1 (Remarks to the Author):

The authors have carried out additional experiments and provide new data to address the comments of the three reviewers. For example, the stability of the compounds has been evaluated. They have also tested additional patient iPSC lines. They have added significant explanation in the body of the manuscript as well. There were some data included in the rebuttal letter that is not in the manuscript. In my opinion, it may help to add the data into the supplementary section.

Response: We appreciate for the helpful comment. To discuss the effect of the 56-59 *in vivo*, we added the data that shows the SOD1-Derlin-1 interaction in the ALS model mice in the **Supplementary Figure 9a**, as the reviewer suggested.

Reviewer #2 (Remarks to the Author):

As stated in the first round of the reviews, this paper presents novel and relevant findings. In addition, substantial amount of work has gone into this manuscript. In the resubmission, authors have improved the manuscript and removed some of the ambiguities that were present before. Importantly, they have answered our experimental questions.

*Among the additional experiments, they have added additional iPSC-derived MN lines and showed that #56 does not have an effect on the survival of control WT lines or mutant TDP43 line. Instead, compound shows a pro-survival effect in 2 out of 3 mutant SOD1 lines. Importantly, the compound shows increased survival *in vivo* in mutant SOD1 mouse model. I recommend this paper for publication in the present form.*

Response: We are grateful to the reviewer for the interest in our manuscript

Reviewer #3 (Remarks to the Author):

*The authors have answered my questions (without providing any additional data), however, the disconnect between the *in vitro* and cell-based data, and the absence of any *in vitro* binding assay (or cellular thermal shift to show target engagement in cells) continues to be of concern. The authors should state in the manuscript text that binding of the molecules to SOD1 could not be detected by SPR. And the statements about*

Given these shortcomings, I am unsure that the conclusions in the manuscript are supported by the data.

there are grammatical errors throughout the manuscript e.g. P.24 line 347, sentence begins with #56 analogue...

Response: We thank the reviewer for the thoughtful comments. As the reviewer suggested we included the statement about the SPR and added the data in the **Supplementary Figure 7a** (p.16 lines 7-8). The grammatical error that the reviewer pointed out was corrected in the revised manuscript (p.25 lines 7-8).